# Ca2-VDM: Efficient Autoregressive Video Diffusion Model with Causal Generation and Cache Sharing

Kaifeng Gao [* 1 2]  Jiaxin Shi [* 3]  Hanwang Zhang [4]  Chunping Wang [5]  Jun Xiao [1]  Long Chen [6]

## Abstract

With the advance of diffusion models, today's video generation has achieved impressive quality. To extend the generation length and facilitate real-world applications, a majority of video diffusion models (VDMs) generate videos in an autoregressive manner, *i.e.*, generating subsequent clips conditioned on the last frame(s) of the previous clip. However, existing autoregressive VDMs are highly inefficient and redundant: The model must re-compute all the conditional frames that are overlapped between adjacent clips. This issue is exacerbated when the conditional frames are extended autoregressively to provide the model with long-term context. In such cases, the computational demands increase significantly (*i.e.*, with a quadratic complexity w.r.t. the autoregression step). In this paper, we propose **Ca2-VDM**, an efficient autoregressive VDM with **Ca**usal generation and **Ca**che sharing. For **causal generation**, it introduces unidirectional feature computation, which ensures that the cache of conditional frames can be precomputed in previous autoregression steps and reused in every subsequent step, eliminating redundant computations. For **cache sharing**, it shares the cache across all denoising steps to avoid the huge cache storage cost. Extensive experiments demonstrated that our Ca2-VDM achieves state-of-the-art quantitative and qualitative video generation results and significantly improves the generation speed. Code is available: https://github.com/Dawn-LX/CausalCache-VDM

---

*Equal contribution [1]Zhejiang University, Hangzhou, China [2]Manycore Tech Inc., Hangzhou, China [3]Xmax.AI Ltd., Beijing, China [4]Nanyang Technological University, Singapore [5]Finvolution Group, Shanghai, China [6]The Hong Kong University of Science and Technology, Hong Kong, China. Correspondence to: Long Chen <longchen@ust.hk>.

*Proceedings of the 42$^{nd}$ International Conference on Machine Learning*, Vancouver, Canada. PMLR 267, 2025. Copyright 2025 by the author(s).

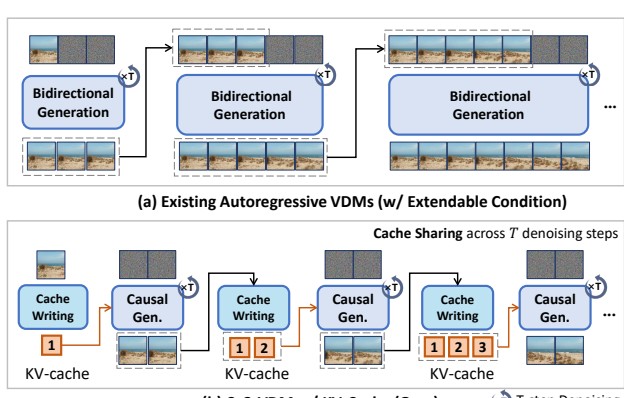

Figure 1: (a): Existing autoregressive VDMs with **bidirectional generation**. The conditional frames can be fixed-length (Henschel et al., 2025; Zheng et al., 2024) or extendable. (b): Our Ca2-VDM, which uses **causal generation** to enable KV-cache and introduce **cache sharing** across all denoising timesteps. **Cache writing** stands for a partial model forward on the denoised frames (*i.e.*, at timestep $t = 0$) until the KV-caches of every layer are computed.

## 1. Introduction

Video diffusion models (VDMs) (Guo et al., 2024b; Ren et al., 2024; Lu et al., 2024; Ma et al., 2025) have made significant advancements by benefiting from the powerful diffusion techniques (Ho et al., 2020; Song et al., 2021a;b) and prior studies on image generation (Rombach et al., 2022; Peebles & Xie, 2023; Chen et al., 2024a). In contrast to images, VDMs need to capture interactions across multiple frames and generate all frames simultaneously (*e.g.*, a 16-frame clip). This is usually facilitated by the temporal attention in prevailing UNet- or Transformer-based VDMs (Wang et al., 2023b; Ma et al., 2025). They introduce interdependencies during the bidirectional attention computation. Consequently, the training and inference lengths must be aligned, extremely restricting the flexibility of VDMs in real-world applications such as long-term (Henschel et al., 2025) or live-stream (Alonso et al., 2024) video generation. Meanwhile, simply scaling the clip length at inference time breaks the alignment and leads to poor generation quality (*e.g.*, Figure 1(b) in (Qiu et al., 2024)), unless

one undertakes time-consuming retraining or fine-tuning.

To address this issue, an effective and prevalent solution is **autoregressive VDMs** (Blattmann et al., 2023a; Henschel et al., 2025; Lu et al., 2024): They are capable of autoregressively generating subsequent clips conditioned on last frames of previous clip, as shown in Figure 1(a). However, the autoregression process of existing VDMs is highly *inefficient and redundant*: The conditional frames constitute the overlapping frames between adjacent autoregression chunks and they are re-computed at each step. This issue is exacerbated when the conditional frames are extended autoregressively to provide the model with long-term context. In such cases, the model must re-compute all the conditional frames concatenated by the previously generated chunks, with a quadratic computational demand w.r.t. the autoregressive step (*cf.* Figure 6 in Sec. 4.3).

To overcome the above limitations, we propose to cache the intermediate features (specifically, the keys and values of every attention layer) at each autoregression (AR) step, and reuse them in subsequent AR steps, as shown in Figure 1(b). In this way, the model 1) eliminates the redundant computations in temporal attention blocks, and 2) reduces the processing length to a constant for other temporal-parallel blocks (*e.g.*, spatial attention and visual-text cross attention) while maintaining the extendable long-term context. To successfully implement the KV-cache in VDMs, two key factors must be carefully considered:

- **Cache Computation**. In existing VDMs, the temporal attention is bidirectional, as shown in Figure 2(a). The frames $z_t^{3,4}$ are denoised conditioned on $z_0^{0,1,2}$, and key/value features of $z_0^{0,1,2}$ are also computed conditioned on $z_t^{3,4}$ at every diffusion timestep $t$ (highlighted by the red box and arrows). It's impossible to precompute and cache the keys and values of $z_0^{0,1,2}$ at previous AR steps, since $z_t^{3,4}$ are not yet available.

- **Cache Storage**. During inference, the VDM is repeatedly called in the denoising process at each AR step, where each call is taken with a different timestep $t$. All most all Existing VDMs (Lu et al., 2024; Ren et al., 2024) use the same timestep embedding (indexed by $t$) for both conditional and noisy frames. This requires each denoising step to have its own cache, *i.e.*, caching the key/value features for all denoising steps will consume huge GPU memory.

In this paper, we propose an efficient autoregressive VDM boosted by causal generation and cache sharing, termed Ca2-VDM, to handle both challenges. For cache computation, we propose **causal generation**: We replace the full temporal attention in each block of the VDM with *causal* temporal attention, and propose *prefix-enhanced* spatial attention. The

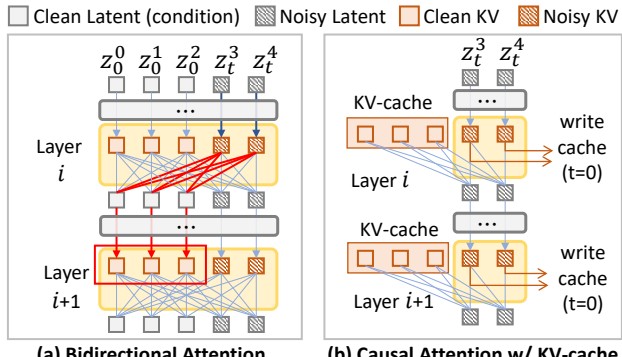

Figure 2: Comparison of bidirectional attention (a) and causal attention (ours) (b). Our design addresses the **cache computation** and **cache storage** issues.

former ensures each generated frame only depends on its prefix frames, and the latter enhances the guidance from the prefix frames. As a result, the cache to be used in subsequent autoregression steps can be precomputed at early steps. For cache storage, we propose **cache sharing**. It leverages the advantages of causal generation: The cache is determined only by the non-noisy preceding (conditional) frames and unaffected by the subsequent noisy frames (*i.e.*, independent of the timestep $t$). Thus, by using a distinct timestep embedding indexed by $t = 0$ for the conditional frames in both training and inference, we enable the cache to be shared across all the denoising steps.

Equipped with causal generation and cache sharing, we propose to store the KV-cache in a queue so that the model can exploit the long-term context while maintaining an affordable computation and storage cost. To support this queue design, the training samples are partially noised to keep clean prefix frames (with random length) as the condition, and the maximum condition length covers the length of KV-cache queue at inference time. Meanwhile, sinusoidal spatial and temporal positional embeddings (*i.e.*, SPEs and TPEs) are added to the frame sequence following Vision Transformer (ViT) (Dosovitskiy et al., 2020). During inference, the TPEs are assigned chunk-by-chunk as the autoregression progresses. To ensure TPEs are correctly assigned when the cumulatively generated video exceeds the training length, we carefully design a cyclic shift mechanism: Cyclic-TPEs [1].

We evaluated our Ca2-VDM on multiple public datasets including MSR-VTT (Xu et al., 2016), UCF-101 (Soomro et al., 2012), and Sky Timelapse (Zhang et al., 2020) for both text-to-video and video prediction tasks. The results

---

[1]Originally, TPEs are re-assigned from scratch at each AR step. However, when KV-cache is enabled, early TPEs have been **bound** to previous KV-caches. They can not be re-assigned (*cf.* Figure 4(c) for more details).

show that our model achieves significant inference speed improvement while maintaining comparable quantitative and qualitative performance as state-of-the-art VDMs. In summary, we make three contributions in this paper: 1) A causal generation structure that allows the intermediate features of conditional frames can be cached and reused in every autoregression step, eliminating the redundant computation. 2) A cache sharing strategy implemented on the KV-cache queue and facilitated by Cyclic-TPEs. It allows the model to acquire extendable context while significantly reducing the storage cost. 3) Our Ca2-VDM achieves comparable performance with SOTA VDMs at a much less computation demand and a high inference speed.

## 2. Related Work

**Video Diffusion Models** (VDMs) have shown impressive generation capabilities, building on the success of latent diffusion models in image generation applications (Rombach et al., 2022; Peebles & Xie, 2023; Chen et al., 2024a). Some works (Lu et al., 2023; Khachatryan et al., 2023; Hong et al., 2023; Zhang et al., 2024) develop training-free methods for zero-shot video generation based on pretrained image diffusion models (*e.g.*, Stable Diffusion (Rombach et al., 2022)). To leverage video training data and improve the generation quality, many works (Ge et al., 2023; Guo et al., 2024b; Wang et al., 2023b; Ren et al., 2024; Dai et al., 2023) extend the 2D Unet in text-to-image diffusion models with temporal attention layers or temporal convolution layers. Recent studies (Ma et al., 2025; Lu et al., 2024) also build VDMs based on spatial-temporal Transformers due to their inherent capability of capturing long-term temporal dependencies. We build our Ca2-VDM based on spatial-temporal Transformers following prior structures.

**Tuning-free Video Extrapolation**. Prior studies have explored autoregressively extrapolating videos using pretrained short video diffusion models without additional finetuning. These methods usually consist of initializing noise sequence based on the DDIM inversion (Song et al., 2021a; Mokady et al., 2023) of previously generated frames (Oh et al., 2024), co-denoising overlapped short clips (Wang et al., 2023a), or iteratively denoising short clips with noise-rescheduling (Qiu et al., 2024). However, their generation quality is upper-bounded by the pretrained VDMs. Meanwhile, the lack of finetuning also leads to temporal inconsistencies between short clip transitions.

**Past-frame Conditioned Video Prediction**. To enhance generation quality and temporal consistency, a popular paradigm is training VDMs conditioned on past frames to predict future frames, enabling video extrapolation through autoregressive model calls. Recent works of autoregressive VDMs have studied a variety of design choices for injecting conditional frames, such as adaptive layer nor-

malization (Voleti et al., 2022; Lu et al., 2024), cross-attention (Zhang et al., 2023b; Lu et al., 2024; Henschel et al., 2025), and explicitly concatenating to the noisy latent along the temporal-axis (Harvey et al., 2022; Lu et al., 2024) or channel-axis (Chen et al., 2024b; Girdhar et al., 2024; Zeng et al., 2024). Some works (Weng et al., 2024; Guo et al., 2024a) also inject conditional frames by adapter-like subnets (*e.g.*, T2I-adapter (Mou et al., 2024) or Control-Net (Zhang et al., 2023a)). In contrast to existing works, our Ca2-VDM avoids the redundant computation of conditional frames by causal generation and cache sharing, and significantly improves the generation speed.

## 3. Method

### 3.1. Preliminaries and Problem Formulation

**Preliminaries**. Diffusion Models (Sohl-Dickstein et al., 2015; Ho et al., 2020) are generative models that model a target distribution $x_0 \sim q(x)$ by learning a denoising process with arbitrary noise levels. To do this, a diffusion process is defined to gradually corrupt $x_0$ with Gaussian noise. Each diffusion step is $q(x_t|x_{t-1}) = \mathcal{N}(x_t; \sqrt{1-\beta_t}x_{t-1}, \beta_t I)$, where $t = 1, \ldots, T$ and $\beta_t \in (0, 1)$ is the variance schedule. By applying the reparameterization trick (Ho et al., 2020), each $x_t$ can be sampled as $x_t = \sqrt{\bar{\alpha}_t}x_0 + \sqrt{1-\bar{\alpha}_t}\epsilon_t$, where $\epsilon_t \sim \mathcal{N}(0, I)$ and $\bar{\alpha}_t = \prod_{i=1}^{t}(1-\beta_i)$. Given the diffusion process, a diffusion model is then trained to approximate the denoising process. Each denoising step is parameterized as $p_\theta(x_{t-1}|x_t) = \mathcal{N}(x_{t-1}; \mu_\theta(x_t, t), \Sigma_\theta(x_t, t))$, where $\theta$ contains learnable parameters.

**Problem Formulation**. Following existing mainstream VDMs (Guo et al., 2024b; Lu et al., 2024; Ma et al., 2025), we develop Ca2-VDM based on latent diffusion models (Rombach et al., 2022) to reduce the modeling complexity of high dimensional visual data. This is achieved by using a pretrained variational autoencoder (VAE) encoder $\mathcal{E}$ to compress $x_0$ into a lower-dimensional latent representation, *i.e.*, $z_0 = \mathcal{E}(x_0)$. Consequently, the diffusion and denoising processes are implemented in the latent space, formulated as $q(z_t|z_{t-1})$ and $p_\theta(z_{t-1}|z_t)$, respectively. The denoised latent $\hat{z}_0$ is decoded back to the pixel space by the pretrained VAE decoder $\mathcal{D}$, *i.e.*, $\hat{x}_0 = \mathcal{D}(\hat{z}_0)$.

In our setting, the model takes as input a VAE encoded latent sequence[2] $z_0^{0:L} = [z_0^0, \ldots, z_0^{L-1}] \in \mathbb{R}^{L \times H \times W \times C}$, where $L$ is the number of frames, $H \times W$ is the downsampled resolution, and $C$ is the number of channels. Then, it aims to generate future frames conditioned on past frames, by learning a distribution $p_\theta(z_0^{P:L}|z_0^{0:P})$. Here the first $P$ prefix frames serve as condition (referred to as **clean prefix**), and the remaining $L - P$ frames are those to be denoised

---

[2]Throughout this paper, we use "$a : b$" to denote a half-open interval ranging from $a$ (inclusive) to $b$ (exclusive)

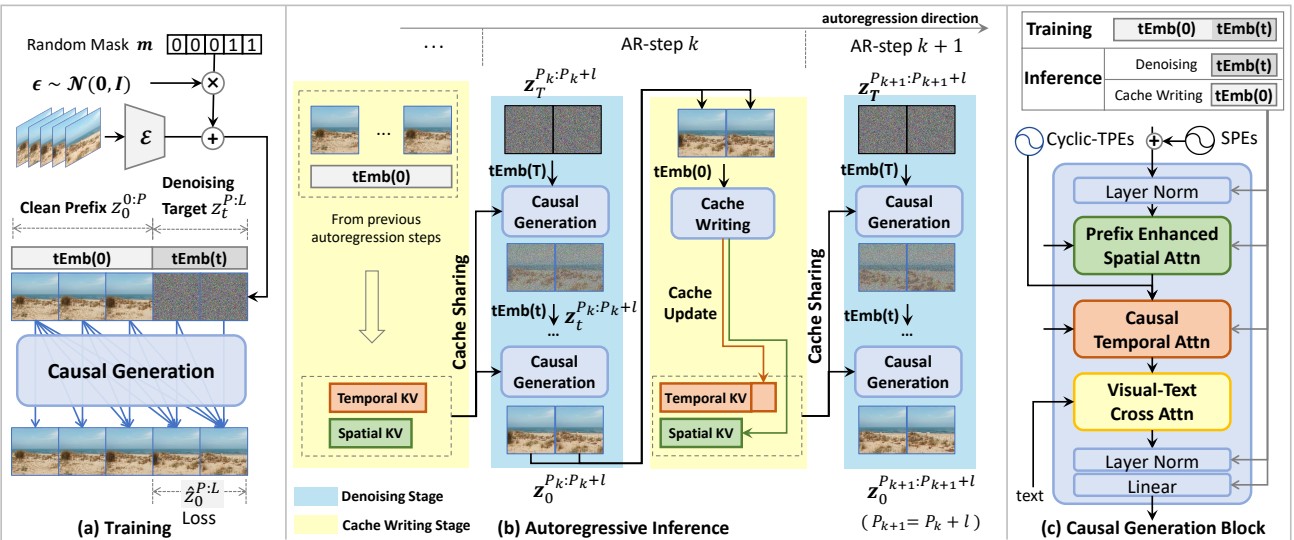

Figure 3: Overview of the Ca2-VDM pipeline. **(a)**: During training, we randomly set $P$ frames clean prefix, and set distinctive timestep embeddings, *i.e.*, **tEmb**$(0)$ for the clean prefix and **tEmb**$(t)$ for the denoising target. **(b)**: During inference, in each autoregression (AR) step, the model denoises an $l$-frame chunk conditioned on the spatial/temporal KV-caches shared across all timesteps (denoising stage), and then computes the keys/values of denoised chunk to update the KV-caches (cache writing stage). **(c)**: Causal generation block. We further illustrate the details of causal temporal attention with Cyclic-TPEs in Figure 4 and the prefix-enhanced spatial attention is left in the Appendix (*cf.* Figure 9).

(referred to as **denoising target**). The model parameterized by $\theta$ is denoted as $\boldsymbol{\epsilon}_\theta(\boldsymbol{z}_t^{0:L}, t)$.

The overall pipeline of Ca2-VDM is shown in Figure 3. We first illustrate the **causal generation** in the training stage (Sec. 3.2), as well as the training objectives. Then, we introduce the KV-cache realization combined with the **cache sharing** mechanism in the autoregressive inference stage (Sec. 3.3), and the queue structure for temporal KV-cache supported by Cyclic-TPEs (*cf.* Figure 4).

### 3.2. Causal Generation and Training Objectives

We first introduce the training objectives, followed by the causal generation block (*cf.* Figure 3(c)). Here we focus on the causal temporal attention and prefix-enhanced spatial attention layers. For the visual-text cross attention, it is widely used in VDMs for text-to-video generation (Rombach et al., 2022; Chen et al., 2024a). And it is optional for pure video prediction (Lu et al., 2024). We refer readers to related works (Chen et al., 2024a) for more details.

**Training Objectives**. Existing diffusion models (Ho et al., 2020; Peebles & Xie, 2023) are trained with the variational lower bound of $\boldsymbol{z}_0$'s log-likelihood, formulated as $\mathcal{L}_{\text{vlb}}(\theta) = -\log p_\theta(\boldsymbol{z}_0|\boldsymbol{z}_1) + \sum_t D_{KL}(q(\boldsymbol{z}_{t-1}|\boldsymbol{z}_t,\boldsymbol{z}_0)\|p_\theta(\boldsymbol{z}_{t-1}|\boldsymbol{z}_t))$, where $D_{KL}$ is determined by the mean $\boldsymbol{\mu}_\theta$ and covariance $\boldsymbol{\Sigma}_\theta$. By re-parameterizing $\boldsymbol{\mu}_\theta$ as a noise prediction network $\boldsymbol{\epsilon}_\theta$ and fixing $\boldsymbol{\Sigma}_\theta$ as a constant variance schedule (Ho et al.,

2020), the model can be trained by a simplified objective:

$$\mathcal{L}_{\text{simple}}(\theta) = \mathop{\mathbb{E}}_{\boldsymbol{z},\boldsymbol{\epsilon},t}\left[\|\boldsymbol{\epsilon}_\theta(\boldsymbol{z}_t, t) - \boldsymbol{\epsilon}\|_2^2\right], \ \boldsymbol{\epsilon} \sim \mathcal{N}(0,1). \quad (1)$$

In our setting, each sample is partially noised. We randomly keep $P$ consecutive frames uncorrupted as the clean prefix, and the remaining frames are treated as the denoising target, as shown in Figure 3(a). We use different timestep embeddings for the clean prefix (*i.e.*, **tEmb**$(0)$) and the denoising target (*i.e.*, **tEmb**$(t)$), rather than a unified timestep embedding for the whole video clip as in many existing VDMs (Lu et al., 2024; Ma et al., 2025). This ensures the cache from the clean prefix can be correctly shared across each denoising timestep $t$ at inference time (since the clean prefix is always assigned with **tEmb**$(0)$). Consequently, the simplified objective function for our model is

$$\widetilde{\mathcal{L}}_{\text{simple}}(\theta) = \mathop{\mathbb{E}}_{\boldsymbol{z},\boldsymbol{\epsilon},t}\left[\|(\boldsymbol{\epsilon}_\theta([\boldsymbol{z}_0^{0:P}, \boldsymbol{z}_t^{P:L}], \boldsymbol{t}) - \boldsymbol{\epsilon}) \odot \boldsymbol{m}\|_2^2\right], \quad (2)$$

where $[\cdot, \cdot]$ stands for concatenation along the temporal axis, and $\boldsymbol{t}$ is the timestep vector with $\boldsymbol{t}_i = t$ if $i \geq P$ else $0$. $\boldsymbol{m} \in \{0,1\}^N$ is a loss mask to exclude the clean prefix part, *i.e.*, with $\boldsymbol{m}_i = 1$ if $i \geq P$ else $0$. In practice, we train the model with learnable covariance $\boldsymbol{\Sigma}_\theta$ by optimizing a combination of $\widetilde{\mathcal{L}}_{\text{simple}}$ and $\mathcal{L}_{\text{vlb}}$ (with the same loss mask) following (Nichol & Dhariwal, 2021; Peebles & Xie, 2023). More details are left in Sec. B.

**Causal Temporal Attention**. To introduce the causality, we mask the attention map to force each frame to only

attend to its preceding frames, as shown in Figure 4(a). Specifically, the input to each layer is first permuted by treating the spatial resolution $H \times W$ as the batch dimension, and then linearly projected to query, key, and value features as $\boldsymbol{Q}, \boldsymbol{K}, \boldsymbol{V} \in \mathbb{R}^{L \times C'}$ (for every spatial grid). The causal attention is computed as

$$\text{CausalAttn}(\boldsymbol{Q}, \boldsymbol{K}, \boldsymbol{V}) = \text{Softmax}\left(\frac{\boldsymbol{Q}\boldsymbol{K}^{\text{T}}}{\sqrt{C'}} + \boldsymbol{M}\right)\boldsymbol{V}, \quad (3)$$

where $\boldsymbol{M} \in \mathbb{R}^{L \times L}$ is a lower triangular attention mask with $\boldsymbol{M}_{i,j} = -\infty$ if $i < j$ else $0$. Note that we only describe one attention head and omit the diffusion step $t$ for brevity.

**Prefix-Enhanced Spatial Attention**. In analogy to causal temporal attention, integrating the clean prefix and denoising target into one attention sequence helps enhance the guidance of conditional information. Inspired by prior works (Hu, 2024; Ren et al., 2024), we do this via spatial-wise concatenation (*cf.* Figure 9 in the Appendix). Let $\boldsymbol{h}_t^{0:L} \in \mathbb{R}^{L \times H \times W \times C'}$ be the hidden input to each layer, where the number of frames $L$ is treated as batch dimension and $H \times W$ is flattened for attention calculation. We take a sub-prefix of length $P'$ and concatenate it to the denoising target. Specifically, for $\boldsymbol{h}_t^i$ from the $i$-th frame, the query is $\bar{\boldsymbol{Q}}(i) = \boldsymbol{W}^Q \boldsymbol{h}_t^i$. The prefix-enhanced key is

$$\bar{\boldsymbol{K}}(i) = \begin{cases} \boldsymbol{W}^K[\boldsymbol{h}_0^{P-P'}; ...; \boldsymbol{h}_0^{P-1}; \boldsymbol{h}_t^i] & \text{if } i \geq P \\ \boldsymbol{W}^K[\boldsymbol{h}_0^i; ...; \boldsymbol{h}_0^i] & \text{if } i < P \end{cases}, \quad (4)$$

where $[\cdot; \cdot]$ stands for concatenation along the spatial dimension, and $\boldsymbol{h}_0^i$ is broadcast by self-repeat $P'$ times for every $i < P$ (*i.e.*, the clean prefix part). We do the same operation to obtain the prefix-enhanced value $\bar{\boldsymbol{V}}$. Consequently, for every frame, the prefix-enhanced spatial attention is computed as $\text{Attention}(\bar{\boldsymbol{Q}}, \bar{\boldsymbol{K}}, \bar{\boldsymbol{V}})$ with an attention map of shape $(HW) \times ((P'+1)HW)$. In practice, $P'$ is relatively small (*e.g.*, $P' = 3$), as the computational cost scales proportionally with $P'$, while adjacent prefix frames tend to exhibit similar appearances. We empirically show that prefix enhancement improves the generation quality (*cf.* Table 4).

### 3.3. Autoregressive Inference with Cache Sharing

We first introduce an overview of the autoregressive inference equipped with cache sharing, as shown in Figure 3(b). Then for each autoregression step, we illustrate the temporal KV-cache queue and cyclic temporal positional embeddings (Cyclic-TPEs) . Finally, we introduce the spatial KV-cache for prefix-enhanced spatial attention.

**Autoregressive Inference**. The model starts from a given first frame and generates an $l$-frame chunk per AR step. Each AR step consists of a denoising stage and a cache writing stage. The spatial and temporal KV-caches are shared across every denoising timestep $t$ (*i.e.*, cache sharing). In the denoising stage, given $P_k$ generated frames

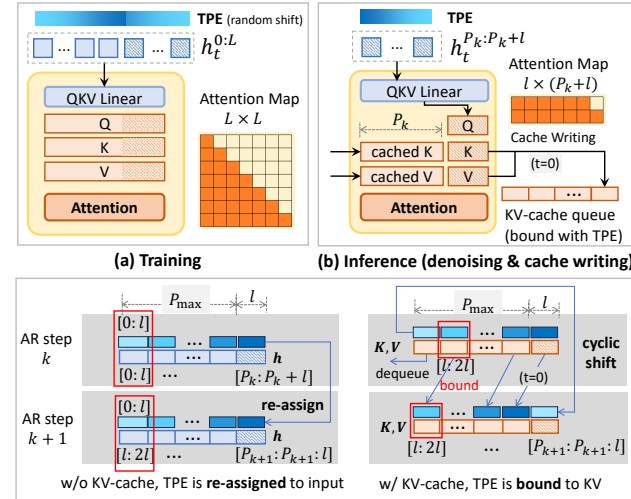

**(a) Training**   **(b) Inference (denoising & cache writing)**

**(c) Original TPEs' reassignment (left) vs. Cyclic-TPEs (right)**

Figure 4: Illustration of causal temporal attention (a) & (b) and the temporal KV-cache queue with Cyclic-TPEs (c). In (c), $L_{\text{train}} = P_{\max} + l$ and $P_{k+l} = P_k + l$. We show the state that autoregressive inference reaches $P_k = P_{\max}$.

at AR step $k$, each denoising step samples $\boldsymbol{z}_{t-1}^{P_k:P_k+l} \sim p_\theta(\boldsymbol{z}_{t-1}^{P_k:P_k+l} | \boldsymbol{z}_t^{P_k:P_k+l}, \boldsymbol{z}_0^{0:P_k})$. Here $\boldsymbol{z}_0^{0:P_k}$ serves as the clean prefix and $\boldsymbol{z}_t^{P_k:P_k+l}$ is the denoising target. Benefiting from the causal generation, the feature computation is uni-directional. This means $\boldsymbol{z}_{t-1}^{P_k:P_k+l}$ is denoised conditioned on $\boldsymbol{z}_0^{0:P_k}$ while the cache of $\boldsymbol{z}_0^{0:P_k}$ could be precomputed in previous autoregression steps without referring to $\boldsymbol{z}_t^{P_k:P_k+l}$. In the cache writing stage, the denoised $\boldsymbol{z}_0^{P_k:P_k+l}$ is input to the model again to compute its *clean* spatial and temporal KV-caches, which will be used in the next AR step.

**Temporal KV-Cache**. Suppose that there are $P_k$ generated frames (*i.e.*, the clean prefix) at AR step $k$. In the denoising stage, the query, key, and value features at timestep $t$ are $\boldsymbol{Q}_t^{P_k:P_k+l}, \boldsymbol{K}_t^{P_k:P_k+l}, \boldsymbol{V}_t^{P_k:P_k+l} \in \mathbb{R}^{l \times C'}$ (considering only one spatial grid). The model reads the *clean* key and value caches as $\boldsymbol{K}_0^{0:P_k}, \boldsymbol{V}_0^{0:P_k} \in \mathbb{R}^{P_k \times C'}$. Then, they are concatenated to the noisy ones as $\tilde{\boldsymbol{K}}(k,t) = [\boldsymbol{K}_0^{0:P_k}, \boldsymbol{K}_t^{P_k:P_k+l}]$ and $\tilde{\boldsymbol{V}}(k,t) = [\boldsymbol{V}_0^{0:P_k}, \boldsymbol{V}_t^{P_k:P_k+l}]$. Finally, the causal temporal attention is computed as:

$$\text{CausalAttn}(\boldsymbol{Q}_t^{P_k:P_k+l}, \tilde{\boldsymbol{K}}(k,t), \tilde{\boldsymbol{V}}(k,t)), \quad (5)$$

where the attention map has a shape of $l \times (P_k + l)$, as shown in Figure 4(b). During denoising, the clean KV-cache $\boldsymbol{K}_0^{0:P_k}$ and $\boldsymbol{V}_0^{0:P_k}$ are shared for every timestep $t$. In the cache writing stage, the clean temporal keys and values are computed as $\boldsymbol{K}_0^{P_k:P_k+l}$ and $\boldsymbol{V}_0^{P_k:P_k+l}$. They are then updated into the KV-cache queue, resulting in $\boldsymbol{K}_0^{0:P_{k+1}}$ and $\boldsymbol{V}_0^{0:P_{k+1}}$, which will be used in AR step $k+1$ (*i.e.*, $P_{k+1} = P_k + l$). As the autoregression progresses, the earliest KV-cache will be dequeued when the length of the

clean prefix $P_k$ reaches a predefined $P_{\max}$ (*i.e.*, a maximum number of conditional frames), as shown in Figure 4(c).

**Cyclic-TPEs**. Assume that the model was trained on video clips with a maximum length of $L_{\text{train}} = P_{\max} + l$ (*i.e.*, with $P_{\max}$ frames clean prefix and $l$ frames denoising target). $L_{\text{train}}$ is also the maximum length of TPE sequence during training. As the autoregressive inference progresses till $P_k = P_{\max}$, the TPEs are used up. When KV-cache is disabled (*cf.* Figure 4(c)-left), to align the training pattern, we can re-assign the TPEs from scratch after the earliest clean frames are dequeued. However, when KV-cache is enabled (*cf.* Figure 4(c)-right), the TPEs were **bound** to keys and values at previous AR steps and had been stored in preceding KV-cache chunks. As a result, we cannot do reassignment to match the training pattern of TPEs. Here we introduce a cyclic shift mechanism, where the denoising target will be assigned those TPEs indexed from the beginning. To support the training/inference alignment of Cyclic-TPEs, in the training stage, each sample is assigned a TPE sequence that is cyclically shifted with a random offset.

**Spatial KV-Cache**. Let $\boldsymbol{h}_t^{P_k:P_k+l}$ be the input to the prefix-enhanced spatial attention at AR step $k$. In the denoising stage, the keys and values from the denoising target are enhanced by the spatial KV-cache (a sub-prefix of $P'$ frames) via spatial-wise concatenation. In the cache writing stage, the denoised latent frames are first enhanced via self-repeat and then computed to obtain the clean spatial keys and values. These operations are aligned with the prefix-enhancement in Eq. (4) of the training stage. Since $P'$ is relatively small ($P' < l$), the prefix enhancement for the current denoising target $\boldsymbol{h}_t^{P_k:P_k+l}$ only depends on spatial KV-cache from the most recent generated chunk (*i.e.*, $\boldsymbol{h}_0^{P_k-l:P_k}$). Thus, in contrast to the queue structure for temporal KV-cache, we only store the spatial KV-cache for one chunk and overwrite it at every AR step.

**Discussion**. It's worth noting that our KV-cache queue for autoregressive VDMs is not a trivial extension of the KV-cache techniques from large language models (LLMs): 1) LLMs predict the next token at each AR step, and the KVs are computed and cached *simultaneously* in each forward call. For VDMs, however, the model is repeatedly called during denoising (with different $t$). This brings the cache computation and storage issues as introduced in Sec. 1. Our implementation solves these two issues, sharing the cache across every denoising step. 2) Caching visual KVs costs much more storage than KVs for text since each token in our setting corresponds to $HW$ visual grids. The queue structure for KV-cache is essential for VDMs considering this heavy storage cost. Early KVs can be safely dequeued as the appearance and motion of new frames are primarily influenced by the most recent KVs. Meanwhile, we propose Cyclic-TPEs to facilitate this mechanism.

# 4. Experiments

## 4.1. Experimental Setup

**Model Details and Baselines**. We built Ca2-VDM based on spatial-temporal Transformer following (Ma et al., 2025; Chen et al., 2024a) and initialized it with Open-Sora v1.0 (Zheng et al., 2024). Following PixArt-$\alpha$ (Chen et al., 2024a), we used T5 (Raffel et al., 2020) as the text encoder and used the VAE from StableDiffusion (Rombach et al., 2022). The length of the clean prefix was randomly sampled according to the multiples of chunk length $l$, *i.e.*, $P \in \{1, 1+l, \ldots, 1+nl\}$ and $P_{\max} = 1+nl$. We used training videos of various lengths with $L_{\text{train}} = P + l$. As comparisons, we built two bidirectional baselines (*cf.* Figure 1(a)) based on the same Open-Sora v1.0: One was trained with fixed-length conditional frames (denoted as OS-Fix), where $P$ is fixed as $P = L_{\text{train}}/2$ in training and inference. The other was trained with autoregressively extendable conditional frames using the same training configs as Ca2-VDM (denoted as OS-Ext).

**Training Details** We conducted training on the text-to-video (T2V) generation and video prediction (*i.e.*, without text prompt) tasks. For T2V generation, we trained OS-Fix and Ca2-VDM on a large-scale video-text dataset InternVid (Wang et al., 2024), by filtering it to a sub-set of 4.9M high-quality video-text pairs. The models were trained video clips at resolution $256 \times 256$ with $l$=16 and $P_{\max} = 1+3l = 49$. For video prediction, we trained OS-Fix, OS-Ext, and Ca2-VDM on the SkyTimelapse (Zhang et al., 2020) dataset at resolution $256 \times 256$ with $l$=8. OS-Ext and Ca2-VDM both used $P_{\max} = 1+3l = 25$. OS-Fix used a fixed $P = 8$. More hyperparameters are left in Sec. C.

**Evaluation Datasets and Metrics**. We used MSR-VTT (Xu et al., 2016), UCF101 (Soomro et al., 2012), and SkyTimelapse (Zhang et al., 2020) datasets at resolution $256 \times 256$, and reported Fréchet Video Distance (FVD) (Unterthiner et al., 2019) following previous works (Zeng et al., 2024; Ge et al., 2023; Chen et al., 2024b). More details about choosing text prompts and computing FVD scores on these datasets are left Sec. D

## 4.2. Evaluation for Generation Quality

We first compared the in-chunk generation quality of Ca2-VDM with SOTA VDMs. Then, we evaluated the temporal consistency of the autoregressive generation. Finally, we conducted ablation studies on Ca2-VDM's design choices.

**In-Chunk Generation Quality**. We evaluated the zero-shot text-to-video (T2V) FVD scores on MSR-VTT (Xu et al., 2016) and UCF101 (Soomro et al., 2012), as shown in Table 1. We compared Ca2-VDM to state-of-the-art T2V models including two groups: 1) Text conditioned: ModelScope (Wang et al., 2023b), VideoComposer (Wang et al.,

Table 1: Zero-shot FVD scores on MSR-VTT (Xu et al., 2016) and UCF101 (Soomro et al., 2012) test sets. All methods generate video at a resolution of $16\times256\times256$. C: condition. T and I are text and image conditions, respectively.

| Method | C | MSR-VTT | UCF101 |
|---|---|---|---|
| ModelScope (Wang et al., 2023b) | T | 550 | 410 |
| VideoComposer (Wang et al., 2023c) | T | 580 | - |
| Video-LDM (Blattmann et al., 2023b) | T | - | 550.6 |
| PYoCo (Ge et al., 2023) | T | - | 355.2 |
| Make-A-Video (Singer et al., 2023) | T | - | 367.2 |
| AnimateAnything (Dai et al., 2023) | T+I | 443 | - |
| PixelDance (Zeng et al., 2024) | T+I | 381 | **242.8** |
| SEINE (Chen et al., 2024b) | T+I | **181** | - |
| Ca2-VDM | T+I | **181** | 277.7 |

Table 2: Finetuned FVD scores on UCF-101 (Soomro et al., 2012) test set. Methods with $*$ were trained on both train and test sets.

| Method | Res. | FVD |
|---|---|---|
| MCVD (Voleti et al., 2022) | $64^2$ | 1143 |
| VDT (Lu et al., 2024) | $64^2$ | 225.7 |
| DIGAN$^*$ (Yu et al., 2022) | $128^2$ | 577 |
| TATS (Ge et al., 2022) | $128^2$ | 420 |
| VideoFusion (Luo et al., 2023) | $128^2$ | 220 |
| LVDM$^*$ (He et al., 2022) | $256^2$ | 372 |
| PVDM (Yu et al., 2023) | $256^2$ | 343.6 |
| Latte (Ma et al., 2025) | $256^2$ | 333.6 |
| Ca2-VDM | $256^2$ | **184.5** |

Table 3: FVD results on MSR-VTT test set.

| Method | FVD between AR step 1 and $i$ | | | | |
|---|---|---|---|---|---|
| | $i=2$ | $i=3$ | $i=4$ | $i=5$ | $i=6$ |
| GenLV | 282.8 | 291.4 | 299.0 | 318.2 | 310.3 |
| StreamT2V | 317.5 | 434.7 | 478.2 | 462.0 | 512.4 |
| OS-Fix | 182.9 | 210.6 | **260.8** | 284.3 | 315.1 |
| Ca2-VDM | **160.6** | **206.5** | 262.8 | **281.3** | **304.7** |

Table 4: Ablations of $P_{max}$ and prefix-enhancement (PE) on SkyTimelapse (Zhang et al., 2020). Each variant of Ca2-VDM generated 48 frames by 6 AR steps. The results were divided into three 16-frame chunks for FVD evaluation.

| $P_{max}$ | PE | Chunk Id | | |
|---|---|---|---|---|
| | | 1 | 2 | 3 |
| 25 | $\times$ | 274.8 | 244.5 | 275.1 |
| 25 | $\checkmark$ | 257.4 | 216.5 | **238.5** |
| 41 | $\times$ | 187.3 | 209.3 | 263.2 |
| 41 | $\checkmark$ | **185.0** | **202.9** | 240.5 |

2023c), Video-LDM (Blattmann et al., 2023b), PYoCO (Ge et al., 2023), and Make-A-Video (Singer et al., 2023). 2) Text with extra image conditioned, *e.g.*, for image-to-video: AnimateAnything (Dai et al., 2023), PixelDance (Zeng et al., 2024) and video transition: SEINE (Chen et al., 2024b). We also finetuned Ca2-VDM on UCF101 at resolution $16\times256\times256$ and reported the FVD scores in Table 2. We compared it with SOTA video generation models: MCVD (Voleti et al., 2022), VDT (Lu et al., 2024), DIGAN (Yu et al., 2022), TATS (Ge et al., 2022), LVDM (He et al., 2022), PVDM (Yu et al., 2023), and Latte (Ma et al., 2025). The FVD results in both Table 1 and Table 2 show that our Ca2-VDM has a competitive T2V performance with SOTA models. More qualitative examples are left in Sec. E.

**Temporal Consistency**. We compared Ca2-VDM with the two baselines (*i.e.*, OS-Fix and OS-Ext) and existing SOTA autoregressive VDMs. To the best of our knowledge, existing autoregressive VDMs all use fixed-length conditional frames (similar to OS-Fix). We used Gen-L-Video (GenLV) (Wang et al., 2023a) and StreamT2V (Henschel et al., 2025). Specifically, GenLV utilizes a base model AnimateDiff (Guo et al., 2024b) and conducts co-denoising for overlapped 16-frame clips. We implemented it with an overlapping length (*i.e.*, the condition length) of 8 frames. StreamT2V is based on Stable Video Diffusion (Blattmann et al., 2023a) and finetunes it conditioned on preceding frames to generate subsequent frames. It also generates 16 frames at each AR step, with 8 frames as the condition.

We evaluated the FVD scores of each autoregression (AR) chunk w.r.t. the first chunk, as shown in Table 3. We can observe that Ca2-VDM has relatively lower FVD scores than

the others. This indicates that extendable (long-term) condition helps to improve the temporal consistency. We also show qualitative examples in Figures 5. It shows content mutations in consecutive frames from the results of fixed-length condition methods, *e.g.*, the $24^{th}$ and $25^{th}$ frames in GenLV, and the $65^{th}$ and $66^{th}$ frames in StreamT2V. We further compared Ca2-VDM with the condition extendable baseline, *i.e.*, OS-Ext (*cf.* Figure 7). We see that Ca2-VDM shows comparable results with OS-Ext (while being more computationally efficient as demonstrated in Sec. 4.3). We conducted further comparisons between Ca2-VDM and OS-Ext in terms of video quality and long-term content drift. The results are left in Sec. E of the Appendix.

**Ablation Studies**. We studied the effectiveness of longer condition length and the prefix-enhancement (PE) in spatial attention (*cf.* Eq. (4)). We trained variants of Ca2-VDM with different $P_{max}$ or without PE. The results are reported in Table 4. Each model was called with 6 AR steps to generate a 49-frame video (with the given first frame) and evaluated by the FVD scores of three 16-frame chunks (exclude the first frame) w.r.t. the 16-frame ground-truth videos. We can see that both increasing $P_{max}$ and using PE are beneficial in improving the generation quality.

### 4.3. Evaluation for Autoregression Efficiency

We evaluated the efficiency in two aspects: 1) time cost for autoregressive generation, and 2) detailed computational costs for each component in the Transformer blocks.

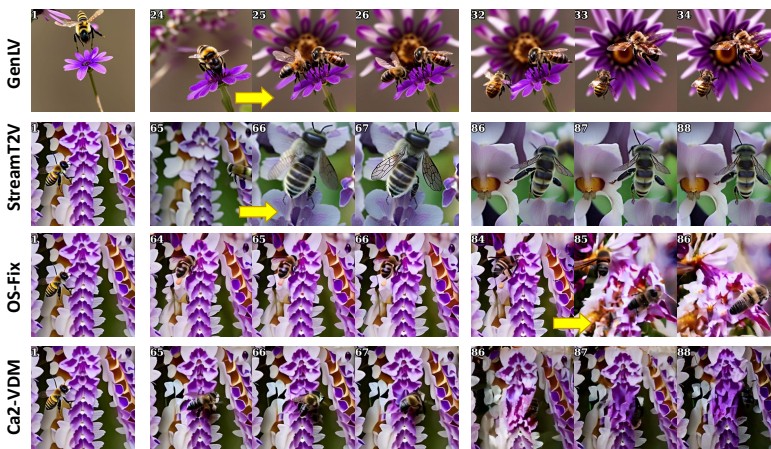

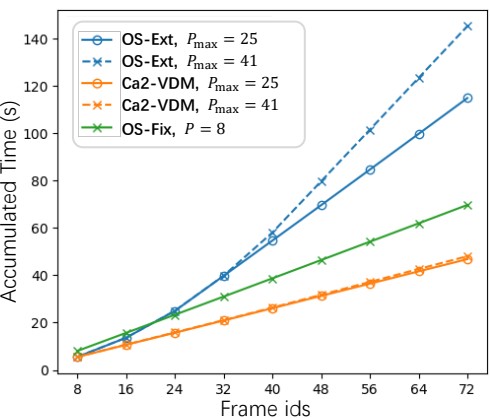

Figure 5: Qualitative examples from GenLV (Wang et al., 2023a), StreamT2V (Henschel et al., 2025), OS-Fix (Zheng et al., 2024), and Ca2-VDM. Yellow arrows highlight consecutive frames having mutations.

Figure 6: Accumulated time cost *w.r.t.* frame ids. We show OS-Ext and Ca2-VDM with $P_{\max} = 25$ and $41$, and OS-Fix with a fixed $P = 8$.

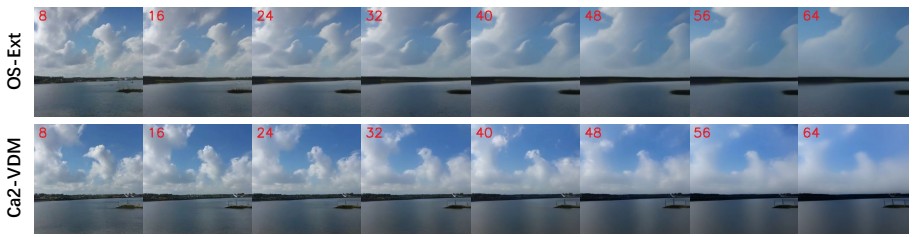

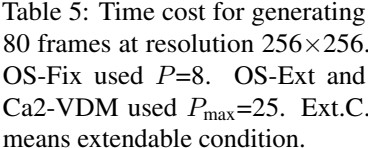

Figure 7: Results from OS-Ext and Ca2-VDM. They have comparable quality, while Ca2-VDM is more computationally efficient, as evidenced in Table 5, Figure 6 and 8.

Table 5: Time cost for generating 80 frames at resolution $256 \times 256$. OS-Fix used $P=8$. OS-Ext and Ca2-VDM used $P_{\max}=25$. Ext.C. means extendable condition.

| Method | Ext.C. | Time (s) |
|---|---|---|
| StreamT2V | | 150 |
| OS-Ext | ✓ | 130.1 |
| OS-Fix | | 77.5 |
| Ca2-VDM | ✓ | **52.1** |

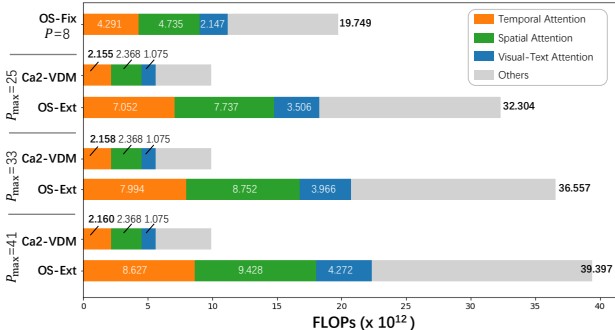

Figure 8: Number of floating-point operations (FLOPs) for generating 56 frames (7 AR steps). All results were computed by conducting only one denoising step for simplicity.

see that Ca2-VDM significantly improved over OS-Fix, OS-Ext, and StreamT2V (Henschel et al., 2025), while being compatible with extendable condition. We further evaluated the accumulated time cost till each AR step, as shown in Figure 6. We can observe that: 1) Compared to OS-Fix, the time cost in Ca2-VDM has a clear reduction since it does not have redundant computations. 2) As the condition extends, the time cost of OS-Ext grows quadratically (before $P_{\max}$ is reached), while the time cost of Ca2-VDM only grows linearly. 3) As the $P_{\max}$ grows to incorporate longer condition, the increase of time cost for OS-Ext is significant, while it is relatively slight for Ca2-VDM.

**Computational Cost**. We counted the floating-point operations (FLOPs) of temporal, spatial, and visual-text attention layers in the Transformer blocks (*cf.* Figure 8). As the $P_{\max}$ grows, the increased computations are seen in all three types of attention layers for OS-Ext. In contrast, for Ca2-VDM, the number of FLOPs only slightly increases in the temporal attention, while keeping constant in other operations. This is because the extended conditional frames only participate in the computation as temporal KV-caches.

**Memory Cost**. We conducted empirical GPU memory

**Time Cost**. We first show the cumulative time cost of autoregressive generation in Table 5. Our models were tested on a single NVIDIA A100 GPU to generate 80 frames at resolution $256 \times 256$, using improved DDPM (Nichol & Dhariwal, 2021) with 100 denoising steps. The result of StreamT2V (Henschel et al., 2025) is from its GitHub page, which was tested on the same device and resolution. We can

Table 6: GPU memory usage comparison between Live2diff (Xing et al., 2024) and Ca2-VDM. The comparisons are not strictly aligned since Live2diff is Unet-based. The resolution of the generated video is $256 \times 256$. $L$ is the number of generated frames at each auto-regression step. $H$ and $W$ are after $8\times$ VAE down sampling. The values of $h'w'$ and $C'$ vary across blocks due to the down-sampling and up-sampling in Unet. PE means prefix-enhancement (*cf.* Eq.(4)).

| Method | Denoising Steps $(T)$ | Model Forward Shape $(B, C, L, H, W)$ | KV-cache Shape $(T, L_{\text{cond}}, hw, C')$ | KV-cache Memory Cost | Total Memory Cost |
|---|---|---|---|---|---|
| Live2diff | 4 | $(4, 4, 1, 32, 32)$ | $(4, 16, h'w', C')$ | 1.42 GB | 10.90 GB |
| Live2diff | 50 | $(50, 4, 1, 32, 32)$ | $(50, 16, h'w', C')$ | 17.70 GB | 29.46 GB |
| Ca2-VDM w/ PE | 50 | $(1, 4, 8, 32, 32)$ | $(1, 25, hw, C)$ | 0.86 GB | 4.79 GB |
| Ca2-VDM w/o PE | 50 | $(1, 4, 8, 32, 32)$ | $(1, 25, hw, C)$ | 0.77 GB | 3.95 GB |

statistics, as shown in Table 6. We compared Ca2-VDM with a concurrent work, Live2diff (Xing et al., 2024). It stores KV-cache for every denoising step (with different noise levels $t$ and thus different KV features), which costs much more GPU memory than ours. Note that Live2diff uses a batch size that is equal to the number of denoising steps, *i.e.*, $B = T$. This is because it uses pipeline denoising following StreamDiffusion (Kodaira et al., 2023), which puts frames with progressive noisy levels into a batch and generates one frame each autoregression step. Benefited from cache sharing, Ca2-VDM's memory cost is independent of denoising steps, as its fixed shape $(1, 25, hw, C)$ ensures constant memory usage. In contrast, Live2diff's memory cost scales with $T$ (*e.g.*, from 1.42 GB at $T = 4$ to 17.70 GB at $T = 50$), confirming that cache sharing saves $T\times$ GPU memory. As a result, Ca2-VDM requires only 0.86 GB (w/ PE) or 0.77 GB (w/o PE), with the difference due to spatial KV-cache for prefix-enhancement (PE).

## 5. Conclusions

In this paper, we present an efficient autoregressive video diffusion model, *i.e.*, Ca2-VDM. It has two key designs: causal generation and cache sharing. The former eliminates the redundant computations of conditional frames. The latter significantly reduces the storage cost. Our model shows comparable generation quality with existing SOTA VDMs with existing bidirectional attention while achieving notable speedup for the autoregressive generation.

## Acknowledgements

This work was supported by the National Key Research & Development Project of China (2024YFB3312900), Key R&D Program of Zhejiang (2025C01128), an Fundamental Research Funds for the Central Universities. Long Chen was supported by the Hong Kong SAR RGC Early Career Scheme (26208924), the National Natural Science Foundation of China Young Scholar Fund (62402408), Huawei Gift Fund, and the HKUST Sports Science and Technology Research Grant (SSTRG24EG04). Kaifeng Gao was supported by the 2024-2025 Grant for Pursuing Outstanding Doctoral Dissertations of Zhejiang University.

## Impact Statement

Our Ca2-VDM is a generic fast video generation paradigm. It is potentially powerful to boost existing VDMs to generate high-quality live-stream videos. The live-stream (or real-time) video generation techniques have a revolutionary impact on the field of content creation industry, and have great potential commercial values. Meanwhile, it's necessary to note that Ca2-VDM also has the inherent risks of common image/video generation models, such as generating videos with harmful or offensive content, or being used by malicious actors for generating fake news. We can use some watermarking technologies (*e.g.*, (Lukas & Kerschbaum, 2023)) to avoid the generated videos being abused.

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

# Appendix

## A. Illustration of Prefix-enhanced Spatial Attention

We provide more details of Prefix-enhanced Spatial Attention (*cf.* Eq. (4)) in Figure 9.

## B. Detailed Training Objectives

Recall that (*cf.* Sec. 3.2 in the main text) existing diffusion models (Ho et al., 2020; Nichol & Dhariwal, 2021; Peebles & Xie, 2023) are trained with the variational lower bound of $z_0$'s log-likelihood, formulated as

$$\mathcal{L}_{\text{vlb}}(\theta) = -\log p_\theta(z_0|z_1) \\ + \sum_t D_{KL}(q(z_{t-1}|z_t, z_0)\|p_\theta(z_{t-1}|z_t)). \quad (6)$$

Since $q$ and $p_\theta$ are both Gaussian, $D_{KL}$ is determined by the mean $\boldsymbol{\mu}_\theta$ and covariance $\boldsymbol{\Sigma}_\theta$. By re-parameterizing $\boldsymbol{\mu}_\theta$ as a noise prediction network $\boldsymbol{\epsilon}_\theta$ and fixing $\boldsymbol{\Sigma}_\theta$ as a constant variance schedule (Ho et al., 2020), the model can be trained using a simplified objective function:

$$\mathcal{L}_{\text{simple}}(\theta) = \mathbb{E}_{z,\epsilon,t} \left[ \|\boldsymbol{\epsilon}_\theta(z_t, t) - \boldsymbol{\epsilon}\|_2^2 \right], \ \boldsymbol{\epsilon} \sim \mathcal{N}(0, 1). \quad (7)$$

In our setting, the simplified objective function is

$$\widetilde{\mathcal{L}}_{\text{simple}}(\theta) = \mathbb{E}_{z,\epsilon,t} \left[ \|(\boldsymbol{\epsilon}_\theta([z_0^{0:P}, z_t^{P:L}], t) - \boldsymbol{\epsilon}) \odot m\|_2^2 \right]. \quad (8)$$

Following prior works (Nichol & Dhariwal, 2021; Peebles & Xie, 2023), we train the model with learnable covariance $\boldsymbol{\Sigma}_\theta$ to improve the sampling quality. This is achieved by optimizing the full $D_{KL}$ term in $\mathcal{L}_{\text{vlb}}$, resulting in an $\widetilde{\mathcal{L}}_{\text{vlb}}$ in our setting, *i.e.*, applied with the same timestep vector $t$ and loss mask $m$. Then, the model is optimized by a combined loss function $\widetilde{\mathcal{L}}_{\text{simple}} + \widetilde{\mathcal{L}}_{\text{vlb}}$.

## C. Training Details and Hyperparameters

**Text-to-Video (T2V) Training**. We trained Ca2-VDM and the OS-Fix baseline on a large-scale video-text dataset InternVid (Wang et al., 2024), by filtering it to a sub-set of

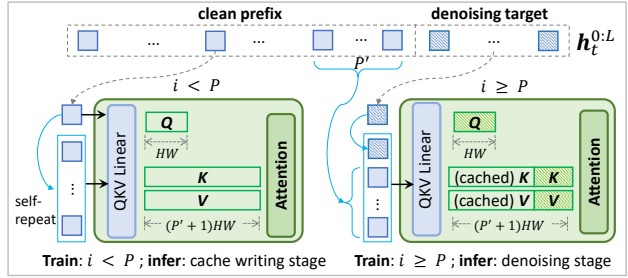

Figure 9: Illustration of prefix-enhanced spatial attention. For $i \geq P$, the left part of $K, V$ is from clean prefix (in training) or cached $K, V$ (in the denoising stage of inference).

4.9M high-quality video-text pairs with resolution $256 \times 256$. For Ca2-VDM, the training consists of two stages. We first train the causal modeling ability without the clean prefix (*i.e.*, without conditional frames) on 32-frame videos. Then we use longer videos of 65 frames to train the model with the clean prefix, *i.e.*, with $l = 16$, $P_{\text{max}} = 1 + 3l = 49$ and $\max(L_{\text{train}}) = P_{\text{max}} + l = 65$. In the first stage, the model was trained with a batch size of 288 for 32k steps. In the second stage, it was trained with a batch size of 144 for 21k steps. For OS-Fix, it was trained with $L_{\text{train}} = 32$ frames and $P = l = L_{\text{train}}/2 = 16$ frames, *i.e.*, the prefix length is fixed. It was trained with a batch size of 288 for 20k steps [3].

**Video Prediction Training**. We trained OS-Fix, OS-Ext, and Ca2-VDM on the SkyTimelapse (Zhang et al., 2020) dataset at resolution $256 \times 256$ with $l = 8$. OS-Ext and Ca2-VDM both used $P_{\text{max}} = 1 + 3l = 25$ (*i.e.*, $L_{\text{train}} = 33$). OS-Fix used a fixed $P = 8$ and $L_{\text{train}} = 16$. All three models were trained with a batch size of 8 for 11k steps [4].

**Hyperparameters**. For all the training, we used the DDPM (Ho et al., 2020) schedule with $T = 1000$, $\beta_1 = 10^{-4}$, and $\beta_T = 0.02$. The models were trained using AdamW (Loshchilov & Hutter, 2019) optimizer with a learning rate of 2e-5. At the inference stage, we used the improved DDPM schedule (Nichol & Dhariwal, 2021) with 100 steps. For text-to-video, we set the classifier-free guidance scale as 7.5.

## D. Evaluation Details

### D.1. Datasets

**MSR-VTT** (Xu et al., 2016). we used its official test split which contains 2990 videos, with 20 manually annotated

---

[3]OS-Fix converges faster than Ca2-VDM since it only needs to learn fixed-length conditional frames.

[4]In contrast to text-to-video, the video prediction task on the SkyTimelapse dataset has less diversity and converges faster. So we used smaller batch size and training steps.

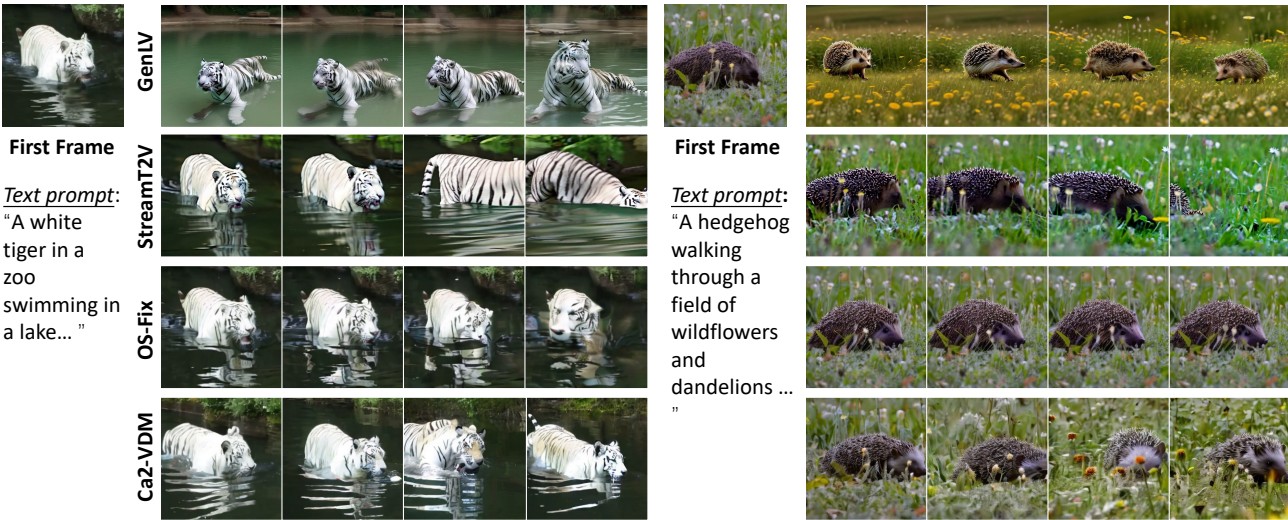

Figure 10: Qualitative examples generated by GenLV (Wang et al., 2023a), StreamT2V (Henschel et al., 2025), OS-Fix, and our Ca2-VDM. We sampled 32 frames with an interval of 8 frames for display. Note that GenLV does not strictly follow the given first frame, since it was not finetuned on explicitly injected conditional frames. In the implementation of GenLV, we used DDIM inversion to build the initial noise based on the first frame.

captions for each video. Following prior works (Ren et al., 2024; Zeng et al., 2024) and for fair comparisons, we randomly selected a caption for each video and generated 2990 videos for evaluation.

**UCF101** (Soomro et al., 2012). As it only contains label names, we employed the descriptive text prompts from PYoCo (Ge et al., 2023), and generated 2048 samples with uniform distribution for each category following (He et al., 2022; Ge et al., 2023; Ren et al., 2024).

**SkyTimelapse** (Zhang et al., 2020). It is a time-lapse dataset showing dynamic sky scenes (*e.g.*, cloudy sky with moving clouds). We used it for video prediction (*i.e.*, without text input). Its training set contains 997 long timelapse videos, which are cut into 2392 short videos. Its test set contains 111 long timelapse videos, which are cut into 225 short videos. We trained the models on its training set and evaluated them on its test set.

### D.2. Quantitative Evaluation

Fréchet Video Distance (FVD) (Unterthiner et al., 2019) measures the similarity between generated and real videos based on the distributions on the feature space. We followed prior works (Blattmann et al., 2023b; Ge et al., 2022; Ren et al., 2024) to use a pretained I3D model[5] to extract the features. We used the codebase[6] from StyleGAN-V (Skorokhodov et al., 2021) to compute FVD statistics.

---

[5] https://github.com/songweige/TATS/blob/main/tats/fvd/i3d_pretrained_400.pt
[6] https://github.com/universome/stylegan-v

For the autoregressive generation results (*e.g.*, the results in Table 3 and Table 4), we calculated the chunk-wise FVD. Specifically, for Table 3, each model generated 48 frames with 6 AR steps and $l = 8$. Since the I3D model accepts at least 16 frames, we evaluated the FVD scores of three 16-frame chunks (*i.e.*, 2 AR steps in each) w.r.t. the 16-frame ground-truth videos. For Table 4, each model generated 96 frames with 6 AR steps and $l = 16$. We evaluated the FVD scores of the generated 16-frame chunk from each AR step w.r.t. the first AR step. Each model generated 512 videos for FVD calculation.

### E. More Experiment Results

In Figure 10 and Figure 11, we show more qualitative examples from GenLV (Wang et al., 2023a), StreamT2V (Henschel et al., 2025), OS-Fix (Zheng et al., 2024), and Ca2-VDM. We can see that Ca2-VDM has comparable generation quality to existing SOTA models.

In Table 7, we evaluated Ca2-VDM and OS-Ext on the VBench (Huang et al., 2024) benchmark. VBench is primarily designed for text-to-video evaluation. For our assessment, we selected four metrics: aesthetic quality, imaging quality, motion smoothness, and temporal flickering. The first two measure spatial (appearance) quality, and the last two assess temporal consistency. The results in Table 7 show that Ca2-VDM achieves comparable performance in both appearance quality and temporal consistency.

In Figure 12, we further compared the long-term content drift (*i.e.*, error accumulation) between Ca2-VDM and the

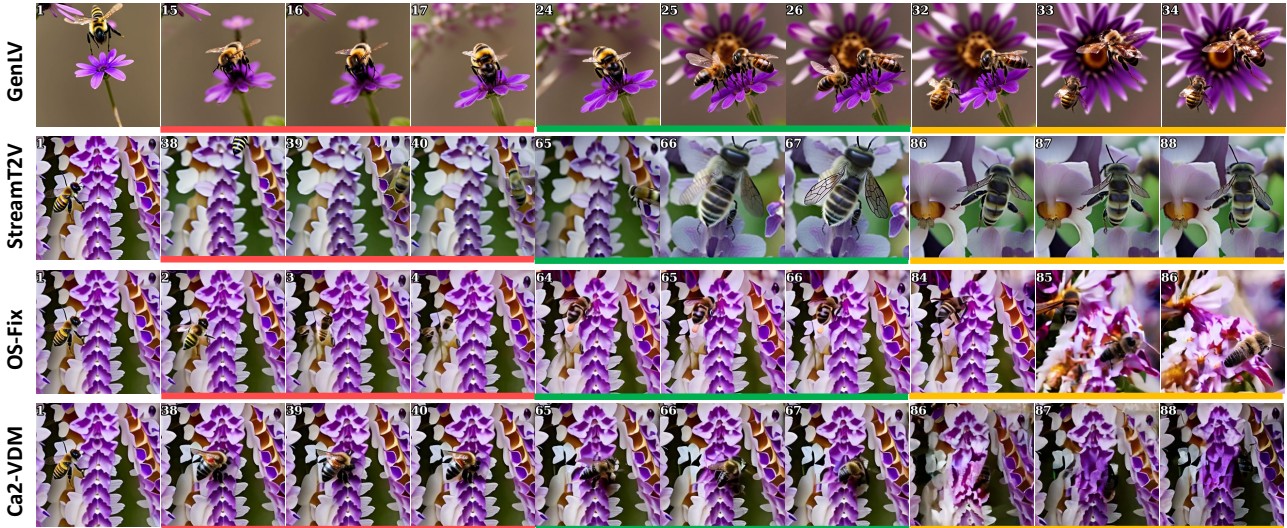

Figure 11: Qualitative examples from GenLV (Wang et al., 2023a), StreamT2V (Henschel et al., 2025), OS-Fix, and our Ca2-VDM. Yellow arrows highlight the consecutive frames having mutations.

Table 7: VBench (Huang et al., 2024) evaluation on Sky-Timelapse (Zhang et al., 2020) test set. The resolution of the generated video is $256 \times 256$. Both models were evaluated with $P_{\max} = 25$ and 6 autoregression steps.

| Method | Aesthetic Quality | Imaging Quality | Motion Smoothness | Temporal Flickering |
|---|---|---|---|---|
| OS-Ext | 44.39 | 50.74 | 98.93 | 98.57 |
| Ca2-VDM | 44.30 | 50.55 | 97.59 | 97.14 |

OS-Ext baseline. As a result, they show comparable visual quality. Both models exhibit a similar degree of error accumulation over time. Given our primary focus on efficiency, we conclude that Ca2-VDM matches the bidirectional baseline while being more efficient in both computation and storage for autoregressive video generation.

## F. Limitations and Possible Future Directions

We analyze the limitations of the current work and propose some possible directions for future work.

**Causal Modeling in Pretraining.** Currently, all the pretrained weights for video diffusion models (either UNet-based, *e.g.*, ModelScore-T2V (Wang et al., 2023b), AnimateDiff (Guo et al., 2024b), or Transformer-based, *e.g.*, Open-Sora (Zheng et al., 2024)) use bidirectional attention in their temporal modules. Our Ca2-VDM is built upon Open-Sora which was also pretrained using bidirectional attention. However, finetuning these bidirectionally pretrained temporal modules using causal attention might be sub-optimal. The weights between bidirectional and causal

temporal attention layers might have inherent gaps. Due to the limited computational resources, we did not conduct causal pretraining. Pretraining the VDM's temporal modules from scratch (using causal attention) might have potential improvements.

**Training Efficiency Trade-off**. Ca2-VDM uses extendable conditional frames and cyclic TPEs. These designs require the model to learn all the possible situations during training. Compared to fixed-length conditional frames and conventional TPEs, the model needs more time to achieve training convergence. Meanwhile, the longer maximum condition length (*i.e.*, $P_{\max}$) we use, the more training is required. On the other hand, once the model is trained, it is more powerful for integrating long-term context. Consequently, it's also potentially beneficial for long-term autoregressive video generation.

**Quality Degradation in Long-term Generation**. As a common challenge, VDMs in long-term autoregressive generation suffer from frame appearance changes and quality degradation. Some works (Henschel et al., 2025; Zhang et al., 2023b) mitigate this issue by providing the VDM with the global appearance information extracted from the initial frame. However, during the long-term generation, video content may change and not all frames commit the same global appearance. In our setting, the long-term extendable context (*i.e.*, early context from the KV-cache queue) helps mitigate the quality degradation, demonstrated by the results in Table 3 and Table 4. Further research on approaches addressing quality degradation is warranted and may hold potential significance for long-term video generation.

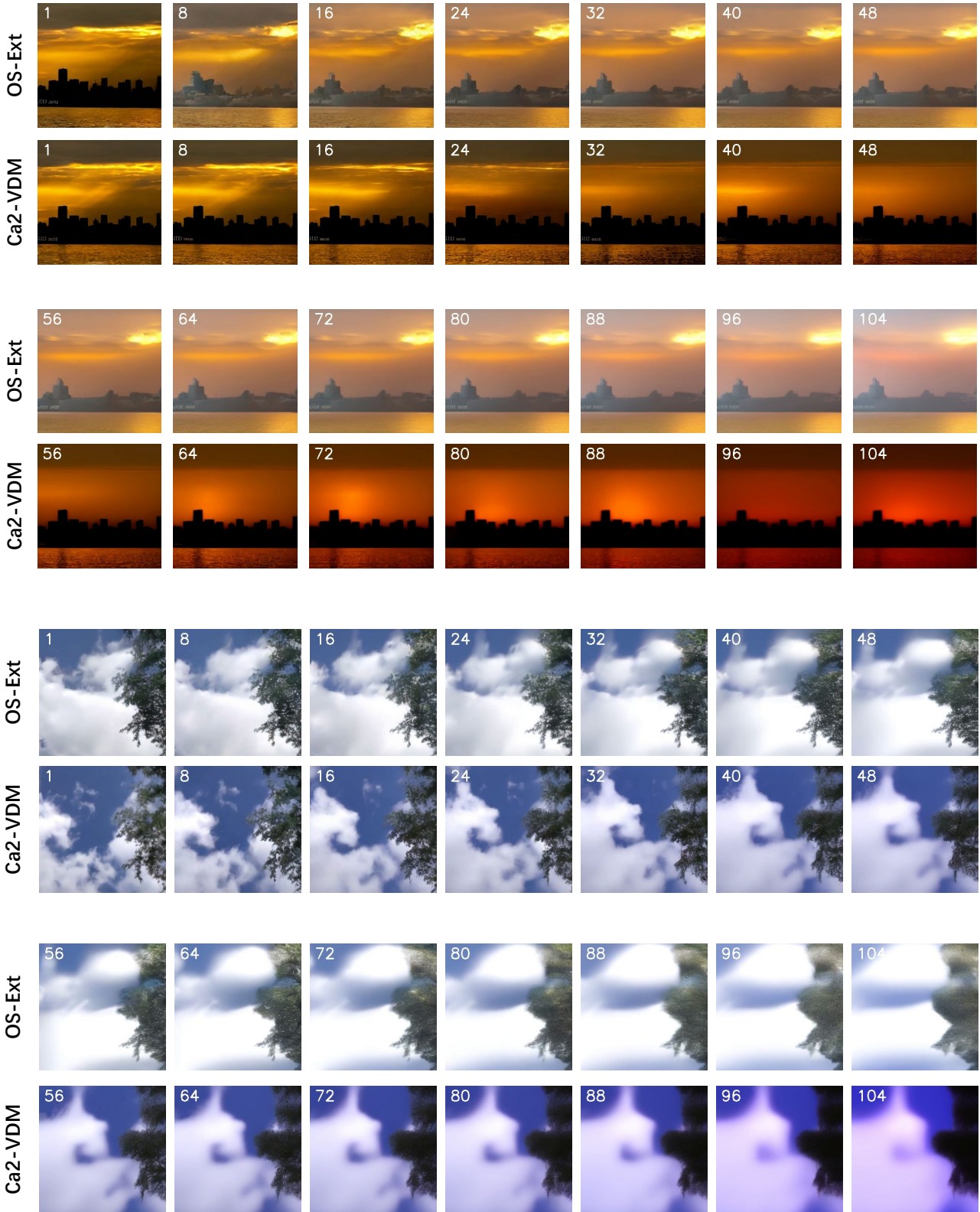

Figure 12: Comparison between OS-Ext and Ca2-VDM in terms of long-term content drift (*i.e.*, long-term quality degradation). Both models were trained on Sky-Timelapse (Zhang et al., 2020). Frame IDs are labeled at top-left corner.

