# OpenReview forum: "Ca2-VDM: Efficient Autoregressive Video Diffusion Model with Causal Generation and Cache Sharing"
_ICML.cc/2025/Conference — ICML 2025 poster_

### Official Review · Reviewer_jMMe · 2025-03-13

**Overall Recommendation:** 2

**Summary:**

This paper introduces a diffusion-based method for video generation via causal transformers. The main idea is to apply kv-caching (technique widely used for AR transformers in NLP) to a causal diffusion transformer. This leads to faster video generation and potentially enables streaming scenarios. The method is based on a OSS text/image2video model (opensora) and results are demonstrated on two video generation benchmarks.

## Update after rebuttal
I am inclined to keep my original score (weak reject). Evaluation is limited, and qualitative results are unsatisfactory (e.g. 19uZ shares this viewpoint).
Similarly, evaluation of the cyclic TPE is not complete: I am not convinced by the argument that comparison to model w/o cyclic TPEs is infeasible - one can probably come up with a smaller-scale experiment to validate it.

**Claims And Evidence:**

- The claim on applicability of the method to long-term or live-stream video generation is not validated.

**Essential References Not Discussed:**

- Auto-regressive diffusion models [Hoogeboom'22] is one of the first works to combine AR and DM into one model.

**Experimental Designs Or Analyses:**

- Validation on longer sequence generation or streaming seems missing.
- Benchmarks are limited.

**Methods And Evaluation Criteria:**

- The set of benchmarks is quite limited (UCF101, MSR-VTT), there are more recent and relatively widely adopted benchmarks: https://github.com/Vchitect/VBench

**Other Comments Or Suggestions:**

- Might be worth clarifying the intro as it reads like temporal attention is inherently bidirectional.
- More results would be useful to understand the quality, on longer sequences in particular.

**Other Strengths And Weaknesses:**

- Clarity: there is a bit of confusing text in the intro about transformers-based VDMs - temporal attention can certainly be causal and must not necessarily be bidirectional - and in practice it is a matter of applying a different mask?
- Quality: The resulting videos do not look like particularly temporally consistent - thus making it hard to understand whether using causal attention (in contrast to dense attention which is more frequently used in recent models such as moviegen).
- Originality: the main contribution of this work is applying auto-regressive modeling and kv-caching which have been widely used in transformer architectures. It is unclear if applying those to VDMs this is a significantly original contribution, especially given that the visuals do not look convincing.
- Limited results: 3 videos seems like not enough to really understand the quality of the model.

**Questions For Authors:**

- One of the issues with the auto-regressive approach is error accumulation. Given that the claim of the paper is ability to generate longer-term videos, did authors try applying their model on a longer context?

**Relation To Broader Scientific Literature:**

This paper is similar in spirit to LVDM-like AR diffusion but applies a common solution for transformers to improve speed.

**Theoretical Claims:**

N/A

---

> ### Author Rebuttal · Authors · 2025-03-31
>
> ## Anonymous link for additional experiment results
>
> https://anonymous.4open.science/r/additional-exp-results-for-anonymous-github-F6EB/readme.md This includes: Table_R1, Table_R2, Figure_R1, and Figure_R2
>
> $~$
>
> ## Q1:  Evaluation on VBench
>
> VBench is primarily designed for text-to-video evaluation. For our assessment, we selected four metrics: aesthetic quality, imaging quality, motion smoothness, and temporal flickering. The first two measure spatial (appearance) quality, and the last two assess temporal consistency. We compared Ca2-VDM and OS-Ext on the Skytimelapse test set.
>
> As shown in Table_R2, Ca2-VDM achieves comparable performance in both appearance quality and temporal consistency. Given our primary focus on efficiency, we conclude that Ca2-VDM matches the bidirectional baseline while being more efficient in both computation and storage for autoregressive video generation.
>
> $~$
>
> ## Q2: The claim on applicability of the method to long-term or live-stream video generation is not validated.
>
> Long-term video generation remains an open problem, with numerous challenges to be solved, including short-term frame quality, long-term content consistency, and generation efficiency.
>
> For our Ca2-VDM, as claimed in the introduction, **the primary focus is to improve the generation efficiency (both computation and storage) of auto-regressive video generation**. Providing superior long-term video generation performance (surpassing SOTAs) is not the primary focus of Ca2-VDM.
>
> For the visual quality, we achieve comparable visual quality to SOTA methods, as evidenced by the FVD results on MSR-VTT and UCF-101 datasets  (Tables 1 and 2), as well as the results on VBench (Table_R2).
>
> For the long-term quality, we conducted additional experiments to show long-term content drift, as in Figure_R1.  In fact, OS-Ext and our Ca2-VDM show comparable visual quality. Both models exhibit a similar degree of error accumulation over time.
>
> $~$
>
> ## Q3: Essential References Not Discussed
>
> >  Hoogeboom, Emiel, et al. "Autoregressive Diffusion Models." in ICLR, 2022
>
> Thank you for the suggested reference. This work proposes Autoregressive Diffusion Model (ARDM), which is built on autoregressive models and incorporates techniques from **discrete diffusion models**. It offers efficient training without the need for causal masking and parallel generation at test time.
>
> In contrast, our Ca2-VDM is built on **continuous latent diffusion models** for video generation, and enables efficient autoregressive generation with causal attention and cache sharing. We will add more discussion in the revised paper.
>
> $~$
>
> ## Q4: Clarity: Confusion in the Introduction about the temporal attention
>
> Thank you for your suggestion. We agree that temporal attention does not necessarily have to be bidirectional. Our introduction aims to highlight that, in current video diffusion models, temporal attention is commonly used and is typically applied in a bidirectional manner, both in UNet and DiT structures. We will improve the writing of the introduction to avoid any potential misleadings in the revised paper.
>
> $~$
>
> ## Q5: Quality: Unsatisfactory temporal consistency, limited qualitative results.
>
> We would like to clarify that the causal attention is not intended to improve the temporal consistency. The causal attention is designed to combine with the kv-cache queue and cache sharing to improve the generation efficiency.
>
> For temporal consistency, as evidenced in Table 3 and Figure 5, our Ca2-VDM offers extendable conditions and achieves better temporal consistency than fixed-length condition SOTA models (StreamT2V, Gen-L-Video, and OS-Fix). We also conducted an additional evaluation in terms of frame-differencing, as shown in Figure_R2. The results show that our method has good temporal consistency, while the other three have periodic content mutations (at the edge of each autoregression chunk), especially GenLV.
>
> Due to limited computational resources and the tight schedule of the rebuttal period, we regret that currently we cannot conduct additional large-scale training to further improve the quality of videos in the supplementary material.
>
> $~$
>
> ## Q6: Originality: Concerns about the contribution
>
> We would like to clarify that applying auto-regressive modeling and kv-cache to VDMs is not trivial (as acknowledged by Reviewer-dKiu).  Our original contribution includes: 1) kv-cache queue boosted with cyclic-TPE and 2) cache sharing
>
> 1. KV-cache queue with cyclic-TPE:   It addresses the cases where positional indices grow beyond the training length while keeping the training-inference alignment. More detailed analysis can also be found in the **Q4 of Reviewer-19uZ**.
> 2. Cache sharing: It enables the model to store only KVs of clean frames, leading to much less GPU memory usage than concurrent work (e.g., Live2diff). More detailed analysis can also be found in the **Q4 of Reviewer-7MWe**.

---

### Official Review · Reviewer_7MWe · 2025-03-14

**Overall Recommendation:** 3

**Summary:**

This work propose an optimized ar video diffusion model, Ca2-VDM, which aims to enhance the efficient long-term, and real-world video generation. In Ca2-VDM, **causal generation** is proposed to reduce the redundant computations of previous conditional frames, and **cache sharing** is proposed to reduce the storage cost during inference. The effectiveness of Ca2-VDM is validated on three different benchmarks, compared with two different conditional generation baselines.

## update after rebuttal

I have read the response from the authors. My concerns are addressed. Some other reviewers are still concerned about the overall poor generation results. I will keep my initial rating.

**Claims And Evidence:**

Yes, the proposed Ca2-VDM is validated by thorough experimental results on different benchmarks.

**Essential References Not Discussed:**

- From Slow Bidirectional to Fast Autoregressive Video Diffusion Models

**Experimental Designs Or Analyses:**

Yes,

**Methods And Evaluation Criteria:**

Yes, the evaluation is the commonly used metrics in generation community like FVD.

**Other Comments Or Suggestions:**

No

**Other Strengths And Weaknesses:**

Strenghts:

1. This work identifies the limitation of existing ar vdms, the repeated computation of conditional frames, which clearly motivate the proposed work to address real-world applications of video generation like long-term video generation and live-stream scenarios.
2. The proposed causal attention and cache sharing is both innovative for long term ar video generation.
3. The paper is well-written, and figures help clarify the proposed method.

**Questions For Authors:**

1. The model is initialized from open-sora v1.0, which is a bidirectional video generation model and there exists mismatch for the causal video generation. Did the authors observe any negative impact or instability during finetuning?
2. One of the causal video generation strengths is that it can be generalized to longer-time video generation, there is no discussion about the this point in this work. The evaluation is conducted under the settings same as the training. Dose the content drift over longer times, like hundreds of frames?
3. The authors claim the cache sharing helps avoid the huge cache storage cost in previous works, but it remains unclear how much actual memory is saved empirically. Could the authors provide detailed metrics of memory footprint comparisons between Ca2-DM and baselines or other methods?
4. The proposed cycle TPEs are interesting contribution, but the paper dose not fully explore its importance separately. Dose the model sensitive to tpe and how much sensitive to the embeddings?

**Relation To Broader Scientific Literature:**

The core contributions of this work, i.e., causal ar temporal generation with cache sharing to significantly improve efficiency of ar video diffusion models, which are deeply related to several lines of research in the broader scientific literature.

**Theoretical Claims:**

No theoretical claims in this work.

---

> ### Author Rebuttal · Authors · 2025-03-31
>
> ## Anonymous link for additional experiment results
>
> https://anonymous.4open.science/r/additional-exp-results-for-anonymous-github-F6EB/readme.md This includes: Table_R1, Table_R2, Figure_R1, and Figure_R2
>
> $~$
>
> ## Q1: Essential References Not Discussed
>
> > Yin, Tianwei, et al. "From slow bidirectional to fast causal video generators." CVPR 2025.
>
> Thank you for your suggestion. This work also introduces causal attention into autoregressive video diffusion models (AR-VDMs). They use distributed matching distillation to align the visual quality of causal student model to the bidirectional teacher model.
>
> This referenced work is a concurrent work with our paper.  I.e., Our initial version of Ca2-VDM was developed concurrently with it.
>
> Our approach distinguishes itself through the kv-cache queue (with cyclic-TPE) and cache sharing mechanisms that provide a both computational and storage efficient framework for (AR-VDMs). We will include this referenced work and add additional discussions in our revised paper.
>
> $~$
>
> ## Q2: Any negative impact or instability observed during finetuning from bidirectional VDM ?
>
> We did not observe any obvious negative impact or instability during our fine-tuning.
>
> Indeed, as verified by the aforementioned reference work (Yin, Tianwei, et al, 2025), distilling the knowledge from a pre-trained bidirectional teacher model to the causal student model brings quality improvement over directly fine-tuning.  Additionally, as discussed in our "Possible Future Directions" section of the Appendix  (Sec. F), pretraining the causal attention from scratch might have potential improvements. We will explore these two strategies in our future work.
>
> $~$
>
> ## Q3: Content drift over longer generated videos
>
> As discussed in our Introduction section, bidirectional models are also capable of generating longer videos beyond the training video length, and this is not a unique strength of causal models. Also, we did not claim long video quality improvement over the bidirectional baseline (OS-Ext) as our contribution.
>
> We conducted additional experiments to show long-term content drift. As shown in Figure_R1, OS-Ext and our Ca2-VDM show comparable visual quality. Both models exhibit a similar degree of error accumulation over time.
>
> Nevertheless, long-term video generation remains an open problem. Both models face challenges in this regard. We can apply the aforementioned distillation technique to enhance the visual quality.
>
> $~$
>
> ## Q4: GPU memory usage between Ca2-DM and baselines
>
> We conducted empirical GPU memory statistics, as shown in Table_R1
>
> We compare Ca2-VDM with a concurrent work, Live2diff [1]. It stores the kv-cache for every denoising step (with different noise level $t$ and thus different KV features) , which costs much more GPU memory than ours. Live2diff uses StreamDiffusion [2]'s pipeline denoising, which puts frames with progressive noisy levels into a batch and generates one frame each autoregression step. So, its batch size in the model forward shape is equal to the denoising steps, i.e., $B=T$.
>
> Note that cache sharing is an inherent property of Ca2-VDM, and it cannot be evaluated without cache sharing. Instead, we demonstrate that Ca2-VDM’s KV-cache memory cost is independent of denoising steps, as its fixed shape $(1,25, hw, C)$ ensures stable memory usage. In contrast, Live2diff's memory scales with $T$ (e.g., from 1.42 GB at $T=4$ to 17.70 GB at $T=50$), confirming that cache sharing saves $T \times$ GPU memory. As a result, Ca2-VDM requires only 0.86 GB (w/ PE) or 0.77 GB (w/o PE), with the difference due to spatial KV-cache for prefix-enhancement (PE).
>
> While Live2diff uses distillation (e.g., LCM) to reduce $T$, existing few-step acceleration methods still require at least 4 steps to generate frames with acceptable quality. This means we can save at least 4 x GPU memory.
>
> In addition,  Live2diff cannot run 50-step denoising with KV-cache when GPU memory is limited or for high-resolution videos. This prevents proper evaluation of the teacher model before distillation, further restricting its applicability.
>
>
>
> [1] Xing, Zhening, et al. Live2diff: Live stream translation via uni-directional attention in video diffusion models. 2024.
>
> [2] Kodaira, Akio, et al. Streamdiffusion: A pipeline-level solution for real-time interactive generation. 2023.
>
> $~$
>
> ## Q5: How much is the model sensitive to cyclic TPEs
>
> The cyclic TPE in Ca2-VDM is specifically designed to enable the model to generate videos that exceed the training length. In other words,  a direct comparison of model performance with and without cyclic TPEs is not feasible. Also, it is not intended to enhance visual quality.
>
> In terms of the model's sensitivity to cyclic TPE, we discussed the impact of training with cyclic TPE in the  "Limitations" of Appendix (Sec. F). It requires the model to learn all the possible situations during training and making it difficult to converge.

---

### Official Review · Reviewer_dKiu · 2025-03-14

**Overall Recommendation:** 3

**Summary:**

Ca2-VDM is an autoregressive video diffusion model designed to generate long videos more efficiently. The paper identifies that existing autoregressive video diffusion models (VDMs) suffer heavy redundant computation when generating videos in chunks: overlapping frames between successive clips are repeatedly processed, leading to quadratic time complexity as more clips are generated​​. To address this, Ca2-VDM introduces two key innovations: (i) a causal generation mechanism with unidirectional temporal attention, so each frame only attends to previous frames. This allows the model to cache intermediate features (keys/values) of past (conditional) frames and reuse them in subsequent generation steps, eliminating redundant computations​. (ii) a cache sharing strategy that uses a fixed-size KV-cache queue along with cyclic temporal positional embeddings (Cyclic-TPEs) to recycle the cache across all diffusion denoising steps​. By sharing cached features rather than storing separate caches per denoising step, memory usage is kept in check. Using these techniques, Ca2-VDM can autoregressively extend video length with only a slight increase in computation per step (linear in the number of steps, rather than quadratic)​. The model is built on a spatial-temporal Transformer (initialized from Open-Sora v1.0, a pre-trained video diffusion model) and is evaluated on both text-to-video generation and video prediction tasks. Experiments show that Ca2-VDM maintains state-of-the-art video generation quality while significantly improving inference speed, enabling generation of longer videos (dozens of frames) more practically​. In summary, the paper’s primary contributions are: (1) a causal generation architecture for VDMs that enables reusing past frame features via caching, (2) a cache-sharing mechanism (with a cyclic positional encoding scheme) that provides long-term context without prohibitive memory growth, and (3) an implementation that achieves comparable or better video quality than prior state-of-the-art models but at much lower computational cost and faster inference​.

**Claims And Evidence:**

The claims in the paper are generally well-supported by empirical evidence. The authors claim that Ca2-VDM eliminates redundant computation and achieves a significant speedup in autoregressive video generation. This is convincingly demonstrated by detailed runtime comparisons. For example, to generate an 80-frame video at 256^2 resolution (with 100 denoising steps), Ca2-VDM required only ~52 seconds on an A100 GPU, whereas a comparable baseline with extendable context (OS-Ext) took 130 seconds and a prior streaming method took 150 seconds​. This confirms a 2.5–3× speedup in practice, matching the claim of improved efficiency. The claim of reduced computational complexity (from quadratic to linear scaling with the number of autoregressive steps) is backed by analysis of FLOPs: as the number of autoregressive steps grows, baseline models see dramatic increases in computation at each new step, while Ca2-VDM’s computation grows only slightly in temporal attention and stays constant in other parts​. The paper also claims state-of-the-art video generation quality (or at least comparable to SOTA). This is supported by quantitative results on standard benchmarks: for text-to-video on MSR-VTT, Ca2-VDM achieves an FVD of 181, matching the best prior model (SEINE) and substantially better than older models (e.g. PixelDance 381, ModelScope 550)​. For video prediction on UCF-101, after fine-tuning, Ca2-VDM attains an FVD of 184.5, outperforming previous methods like VDT (225.7) and VideoFusion (220) by a large margin​. These results substantiate the claim that efficiency was achieved without sacrificing quality. In fact, Ca2-VDM often slightly improves quality in long video settings due to its extended temporal context: the paper shows it yields lower FVD between early and later video chunks compared to baselines, indicating better temporal consistency over long generations​. The qualitative examples (Figure 7) further illustrate that Ca2-VDM avoids the serious frame discontinuities (“mutations” between consecutive frames) that occur in other methods at clip boundaries​. All these observations give clear and convincing evidence for the paper’s claims.

**Essential References Not Discussed:**

N/A

**Experimental Designs Or Analyses:**

The experimental design in this paper is solid and thorough. The authors conduct experiments on multiple datasets covering different scenarios, which strengthens the validity of their findings. For text-conditioned video generation, they evaluate on MSR-VTT (a large benchmark with diverse videos and captions) and UCF-101 (using class names or prompts from prior work for text, ensuring a broad range of actions)​. For unconditional video prediction, they use the Sky Timelapse dataset​, which tests the model’s ability to continue a given video. These choices cover both open-domain content (MSR-VTT) and structured motion (Sky Timelapse’s moving skies), demonstrating the model’s generality. The experiments are designed to isolate the effects of the proposed innovations. Specifically, they compare Ca2-VDM against two controlled baselines built on the same architecture: OS-Fix (which uses fixed-length context like conventional models) and OS-Ext (which allows extended context but without caching, akin to a naive autoregressive extension)​. This is a critical comparison because it shows what benefits come purely from the new design. Indeed, Ca2-VDM vs. OS-Ext reveals the efficiency gains at similar quality, and Ca2-VDM vs. OS-Fix reveals the benefit of having extendable context. They also compare to external baselines like Gen-L-Video (a tuning-free method using overlapping denoising) and StreamingT2V (another autoregressive diffusion approach), which demonstrates where Ca2-VDM stands relative to contemporary solutions​. The inclusion of these baselines indicates a conscientious experimental design aimed at covering all relevant alternatives. The analysis of results is detailed and convincing. The paper reports quantitative metrics (FVD scores) for each dataset and method, and breaks down results by scenario (zero-shot vs. fine-tuned) to ensure fairness​. Notably, the authors include an ablation study (Table 3) on the Sky Timelapse video prediction task, examining the effect of the prefix-enhanced spatial attention and different maximum prefix lengths​. This ablation shows, for instance, that using a longer prefix (more past frames) and the prefix enhancement yields better FVD in later chunks of the video, validating those design choices. They also specifically analyze temporal consistency using their chunk-wise FVD metric (Table 4)​: this analysis revealed that methods with fixed context (OS-Fix, GenLV) or naive streaming can accumulate error and cause noticeable distribution shift in later frames (higher FVD against the first chunk), whereas Ca2-VDM (with extended context) maintains lower drift​. Such an analysis directly addresses the long-term quality claim of the paper. Additionally, the authors present runtime analysis: a table of cumulative inference time per autoregressive step and a FLOPs breakdown by component​. The time analysis (also visualized in Figure 10) clearly demonstrates the linear vs. quadratic growth of time cost, which is central to the paper’s thesis​. It’s commendable that they tested on the same hardware and settings for all methods, even quoting StreamingT2V’s numbers from its GitHub under the same conditions​ – this lends credibility to the comparisons. Overall, the experiments are sound and well-controlled. The paper uses large sample sizes (e.g., generating 512 videos for each condition in consistency tests) to ensure statistical reliability​. Where direct comparison wasn’t feasible (e.g., some baselines not available for certain datasets), the authors either cite published results or reasonably adapt the methods. No obvious flaws or missing analyses were noted. One might have wanted to see a qualitative user study or per-frame perceptual scores, but given the consistency of FVD and the provided visuals, the conclusions seem trustworthy. The only slight issue is that memory usage (GPU memory) for caching vs. not caching is not quantified in the main text – the authors qualitatively claim huge memory savings due to cache sharing​, which is logical. It could have been interesting if they had reported actual VRAM usage for generating long videos with and without cache sharing. However, this omission does not undermine the results; the focus was clearly on computation time, which they addressed thoroughly. In summary, the experimental design is comprehensive, and the analyses directly support the paper’s conclusions without glaring omissions or confounding factors.

**Methods And Evaluation Criteria:**

The proposed methods are well-chosen for the problem of long video generation. The use of causal (unidirectional) temporal attention is a natural solution to avoid re-computation: it’s analogous to how Transformer language models cache past token embeddings to generate long sequences efficiently. Adapting this idea to video diffusion is appropriate and non-trivial – the authors had to modify the video UNet/Transformer to mask temporal attention so that each frame attends only to previous frames​. This ensures that features for past frames (the “conditional” context frames) can be computed once and reused. The introduction of a KV-cache queue for those features, combined with cyclic temporal positional embeddings, is an effective architectural innovation to handle unlimited sequence length. The cyclic positional encoding scheme addresses the fact that positional indices will grow beyond the model’s training length; by wrapping around and randomizing the offset during training, the model learns to handle very long sequences without divergence​. This is an appropriate solution to maintain alignment between training and inference when extending to longer video than seen during training. Additionally, the paper introduces a “prefix-enhanced spatial attention” (a method to feed the past frames’ information into spatial attention layers) – this is a sensible design to strengthen temporal continuity, and the ablation study confirms it modestly improves performance​. Overall, the methodology directly tackles the identified inefficiencies in a principled way and is well-grounded in Transformer design practices. The evaluation criteria and experiments are appropriate for the application domain. The authors evaluate on standard video generation benchmarks for both text-to-video (MSR-VTT, UCF-101 with text prompts) and unconditional video prediction (Sky Timelapse for extrapolating future frames) – covering both major use cases. They use FVD as the primary quantitative metric, which is standard for generative video quality, and they follow common practice by computing FVD with a pre-trained I3D model​. This choice is appropriate, as FVD captures both visual fidelity and temporal coherence. The paper also reports inference time and breakdowns of computation (FLOPs) to support the efficiency claims, which are crucial given the paper’s focus. The combination of quality metrics (FVD), speed measurements, and qualitative examples covers the relevant evaluation angles for a generative model. One could argue for additional metrics like text-to-video relevance (e.g., CLIP score for text alignment) or human evaluation for visual quality, but FVD and the provided visuals are generally accepted indicators for this task. The evaluation protocol is fair: Ca2-VDM is compared against strong baselines, including existing SOTA models (e.g. ModelScope, VideoFusion, SEINE) and carefully constructed ablations (OS-Fix, OS-Ext) that isolate the impact of their contributions. The authors generate a sufficient number of samples (e.g., 2990 videos for MSR-VTT, 2048 for UCF101) for reliable FVD estimation​, and even evaluate temporal consistency by measuring FVD across chunks within a generated video​, which is a thoughtful evaluation of long-term quality. In summary, the methods are well-suited to the problem and the evaluation methodology is rigorous and appropriate for validating both the performance and the efficiency of the proposed approach.

**Other Comments Or Suggestions:**

N/A

**Other Strengths And Weaknesses:**

The biggest weakness of the paper is its scale. This is inevitable considering not all researchers have sufficient GPU resources, but one always may question the results, especially the videos authors provided in the supplementary material --- they are not really convincing, firstly there are only four of them, secondly they all appear blurry and have significant artifacts. Quantitatively, the paper is sound and complete, but these qualitative results do not convince me.

**Questions For Authors:**

N/A

**Relation To Broader Scientific Literature:**

This work is well-situated in the context of the broader literature on video generation and diffusion models.

**Theoretical Claims:**

The paper does not heavily focus on new theoretical derivations。

---

> ### Author Rebuttal · Authors · 2025-04-01
>
> ## Anonymous link for additional experiment results
>
> https://anonymous.4open.science/r/additional-exp-results-for-anonymous-github-F6EB/readme.md This includes: Table_R1, Table_R2, Figure_R1, and Figure_R2
>
> $~$
>
> ## Q1: Additional metrics for per-frame perceptual evaluation
>
> We conducted additional evaluations on the VBench (https://github.com/Vchitect/VBench) benchmark. It is primarily designed for text-to-video evaluation. For our assessment, we selected four metrics: aesthetic quality, imaging quality, motion smoothness, and temporal flickering. The first two measure spatial (appearance) quality, and the last two assess temporal consistency. We compared Ca2-VDM and OS-Ext on the Skytimelapse test set,  as shown in Table_R2.
>
> The results show that Ca2-VDM achieves comparable performance in both appearance quality and temporal consistency. Given our primary focus on efficiency, we conclude that Ca2-VDM matches the bidirectional baseline while being more efficient in both computation and storage for autoregressive video generation.
>
> $~$
>
> ## Additional evaluation on temporal consistency
>
> We also conducted an additional evaluation in terms of frame-differencing, as shown in Figure_R2. The results show that our method has good temporal consistency, while the other three have periodic content mutations (at the edge of each autoregression chunk), especially GenLV.
>
> $~$
>
> ## Q2: GPU memory usage for w/ cache sharing vs. w/o cache sharing
>
>
>
> Cache sharing is integral to Ca2-VDM due to the clean prefix condition design, meaning it cannot be evaluated without cache sharing. Instead, we compared Ca2-VDM with a concurrent work, Live2diff [1], which also uses kv-cache during autoregressive video generation. We conducted empirical GPU memory statistics, as shown in Table_R1
>
> Live2diff stores the kv-cache for every denoising step (with different noise level $t$ and thus different KV features) , which costs much more GPU memory than ours. Live2diff uses StreamDiffusion [2]'s pipeline denoising, which puts frames with progressive noisy levels into a batch and generates one frame each autoregression step. So, its batch size in the model forward shape is equal to the denoising steps, i.e., $B=T$.
>
> Benefited from cache sharing, Ca2-VDM’s KV-cache memory cost is independent of denoising steps, as its fixed shape $(1,25, hw, C)$ ensures constant memory usage. In contrast, Live2diff's memory scales with $T$ (e.g., from 1.42 GB at $T=4$ to 17.70 GB at $T=50$), confirming that **cache sharing saves $T \times$ GPU memory.** As a result, Ca2-VDM requires only 0.86 GB (w/ PE) or 0.77 GB (w/o PE), with the difference due to spatial KV-cache for prefix-enhancement (PE).
>
> While Live2diff uses distillation (e.g., LCM) to reduce $T$, existing few-step acceleration methods still require at least 4 steps to generate frames with acceptable quality. This means we can save at least 4 x GPU memory.
>
> In addition,  Live2diff cannot run 50-step denoising with KV-cache when GPU memory is limited or for high-resolution videos. This prevents proper evaluation of the teacher model before distillation, further restricting its applicability.
>
>
>
> [1] Xing, Zhening, et al. Live2diff: Live stream translation via uni-directional attention in video diffusion models. 2024.
>
> [2] Kodaira, Akio, et al. Streamdiffusion: A pipeline-level solution for real-time interactive generation. 2023.
>
>
> $~$
>
> ## Q3: Unsatisfactory qualitative results of supplementary videos.
>
> We acknowledge that the current qualitative results are not ideal. However, our primary focus is on improving the generation efficiency of video diffusion models, providing a computational and storage efficient framework. The visual quality improvement over the bidirectional attention baseline (OS-Ext) is not claimed as our contribution.
>
> Due to limited computational resources and the tight schedule of the rebuttal period, we regret that we could not conduct additional large-scale training to further improve the qualitative results in the supplementary material. Instead, we conducted experiments on the Sky-Timelapse dataset, as shown in Figure_R1. We can observe that Ca2-VDM achieves comparable qualitative performance to OS-Ext, both in terms of single-frame quality and long-term visual content drift (with a similar degree of error accumulation during long-term video generation).
>
> Nevertheless, we can still apply distillation techniques to improve the quality, e.g., distilling from a bidirectional teacher model to enhance the generation results of our causal generation model (as the student). Additionally, as discussed in the "Future Directions" section of the Appendix (Sec. F), pretraining the causal attention from scratch might have potential improvements.

---

### Official Review · Reviewer_19uZ · 2025-03-16

**Overall Recommendation:** 3

**Summary:**

This paper introduces Ca2-VDM, an autoregressive video diffusion model tailored for generating long videos efficiently. The core idea is to eliminate redundant computation of conditional (overlapped) frames when chaining multiple short clips. To achieve this, the model applies causal generation that replaces standard (bidirectional) temporal attention layers with causal temporal attention and cache sharing that implements a queue-like caching system that holds key/value features from prior frames. Empirical results on text-to-video and video prediction tasks (e.g., MSR-VTT, UCF-101, SkyTimelapse) show that Ca2-VDM achieves SOTA generation quality.

**Claims And Evidence:**

Overall, the evidence is consistent with the main claims. The ideas and methods are straightforward to understand.

**Essential References Not Discussed:**

It might be beneficial to include more discussion of modular approaches (e.g., “ControlNet” or “T2I-Adapter” style methods adapted to video) if relevant for controlling domain shift.

**Experimental Designs Or Analyses:**

The primary design tests text-to-video (MSR-VTT, UCF-101) and unconditional video prediction (SkyTimelapse). The experimental designs are convincing. The analysis is straightforward and aligns well with prior video diffusion literature. The chosen metrics (FVD, speed) are standard for generation tasks. Repeated runs or more diverse metrics (like SSIM or user study) could add to robustness, but the chosen metrics are acceptable.

**Methods And Evaluation Criteria:**

The proposed approach makes sense for scenarios where longer or continuous video generation is needed.

**Other Comments Or Suggestions:**

Further exploration into concurrency or distributed generation strategies might be helpful, especially for generating extremely long or high-resolution videos.

More user-centric evaluations (e.g., user preference tests or other human metrics) could highlight how visually consistent transitions are perceived.

**Other Strengths And Weaknesses:**

Strengths:

1. The causal generation approach plus key/value caching substantially reduce time and GPU cost for longer sequences.
2. Extending the conditional prefix in an autoregressive way often yields more stable transitions between chunks.
3. Multiple standard datasets, thorough runtime analysis, and FVD-based evaluation.

Weaknesses: The paper is difficult to follow. The writing needs significant improvement. While I appreciate the authors’ efforts in implementing and evaluating Ca2-VDM, The design appears to be ad-hoc/incremental to existing techniques.

**Questions For Authors:**

Cache sharing is a popular design for video generation framework, could you highlight/discuss the differences between existing works?

**Relation To Broader Scientific Literature:**

This work is situated in the line of latent diffusion models (Rombach et al., 2022) and follows expansions to video (e.g., Imagen Video, Make-A-Video, LVDM, and so on). It closely resembles other methods that attempt to handle long video generation by chaining short segments (e.g., Gen-L-Video, StreamT2V).

**Theoretical Claims:**

No particularly novel theoretical results (in terms of closed-form proofs or new sampling theorems) are emphasized. The standard diffusion framework is used (Ho et al., 2020 style training objectives; re-parameterization with 𝜖, 𝜃, ϵ, θ), and the model architecture reconfigures attention mechanisms in a causal manner.

---

> ### Author Rebuttal · Authors · 2025-03-31
>
> ## Anonymous link for additional experiment results
>
> https://anonymous.4open.science/r/additional-exp-results-for-anonymous-github-F6EB/readme.md This includes: Table_R1, Table_R2, Figure_R1, and Figure_R2
>
> $~$
>
> ## Q1: Unsatisfactory supplementary video quality
>
> We acknowledge that the current qualitative results are not ideal. However, our primary focus is on improving the generation efficiency of VDMs, providing a computational and storage efficient framework.
>
> Due to limited computational resources and the tight schedule of the rebuttal period, we regret that we can not conduct additional large-scale training to further improve the qualitative results in the supplementary material. Instead, we conducted experiments on SkyTimelapse, as in Figure_R1. It shows that Ca2-VDM achieves comparable qualitative performance to OS-Ext, both in terms of single-frame quality and long-term visual content drift (with a similar degree of error accumulation).
>
> In addition, we can still apply distillation techniques to improve the quality, e.g., distilling from a bidirectional teacher model to enhance the visual quality of Ca2-VDM (as the student). Also, as discussed in the "Future Directions" of Appendix (Sec. F), pretraining the causal attention from scratch might have potential improvements.
>
>
> $~$
>
>
>
> ## Q2: Essential References Not Discussed
>
> > Modular approaches (ControlNet or T2I-Adapter style methods) for domain shift
>
> ControlNet and T2I-Adapter are methods designed to introduce additional conditional signals to VDMs. They provide structural guidance like edges, depth maps, or segmentation maps.
>
> However, our work does not focus on domain adaptation or structure-guided generation. Instead, we generate future frames autoregressively conditioned on previous frames, rather than conditioning on external guidance to synthesize realistic frames from sketches or animate realistic videos.
>
>
> $~$
>
>
>
> ## Q3: Highlight the differences between existing works about cache sharing
>
> In our work, "cache sharing" specifically refers to sharing kv-cache across denoising steps, not for "sharing the cache across autoregressive steps". **To this end, we are the first to introduce cache sharing to autoregressive VDMs.**
>
> To the best of our knowledge, there is also a concurrent work Live2diff [1] that introduces kv-cache to autoregressive VDMs. However, it stores kv-cache for every denoising step (with different noise level $t$ and thus different KV features), which costs much more GPU memory than ours.
>
> In contrast, Ca2-VDM enables cache sharing and only stores the KVs of clean frames. This is boosted by our "clean prefix" conditional frames,  the corresponding timestep sampling strategy, and the training objectives (cf. Eq.2).
>
> We further provide GPU memory comparison in Table_R1. More detailed analysis can be found in the Q4 of Reviewer-7MWe.
>
>
> $~$
>
>
> ## Q4: The design of Ca2-VDM compared to existing techniques
>
> We would like to clarify our original contribution. Our model is the first one to introduce 1) kv-cache queue boosted with cyclic-TPE and 2) cache sharing to VDMs.
>
> 1. KV-cache queue with cyclic-TPE:  It's non-trivial to apply LLMs' kv-cache to VDMs. In LLMs, all tokens from the beginning are maintained during all autoregression steps. However, in VDMs, early conditional frames can be (or should be) removed due to the large memory cost for visual data, as the appearance and motion of new frames are primarily influenced by the most recent frames. Due to the kv-cache queue, early TPEs have been bound to previous KV-caches and can not be reassigned (as discussed in Figure 4(c)). This motivates us to propose Cyclic TPEs to keep training-inference alignment while enabling generation beyond the training length.
>
> 2. Cache sharing: As answered in Q3.
>
>
> $~$
>
>
> ## Q5: Exploration on distributed generation strategies
>
> Thank you for your suggestion. We surveyed some related works. E.g., Video-Infinity [2] offers a distributed approach across multiple devices, where spatial modules operate independently, whereas temporal modules synchronize context via collaborative communications.
>
> We plan to incorporate this kind of method into Ca2-VDM as future work to improve the scalability of video generation.
>
>
> $~$
>
>
>
> ## Q6: More evaluations for visually consistent transitions
>
> We evaluated Ca2-VDM and OS-Ext on the Skytimelapse test set using VBench evaluation, as shown in Table_R2.  The last two metrics (motion smoothness and temporal flickering) measure consistent transitions. It shows that Ca2-VDM achieves comparable performance with OS-Ext.
>
>
> $~$
>
>
>
> ## Q7: Writing improvement
>
> Thank you for your suggestion. We will revise the paper to improve the writing and incorporate the discussions and clarifications in the above answers.
>
> $~$
>
> [1] Xing, Zhening, et al. Live2diff: Live stream translation via uni-directional attention in video diffusion models. 2024.
>
> [2] Tan, Zhenxiong, et al. Video-infinity: Distributed long video generation.  2024.

---

### Decision · Program_Chairs · 2025-05-01

**Decision:**

Accept (poster)

**Comment:**

3x weak accept, 1x weak reject. This paper introduces an autoregressive video diffusion model that leverages causal generation and cache sharing to improve efficiency in long video generation. The reviewers agree on the (1) innovative causal generation mechanism with cache sharing that substantially reduces redundant computation and GPU cost, (2) thorough experimental evaluation on standard benchmarks demonstrating competitive video quality alongside significant speedup, and (3) clear potential for practical applications in extended video generation scenarios. However, they note the (1) clarity and writing issues that render the paper difficult to follow, (2) unsatisfactory and limited qualitative results in the supplementary material, and (3) concerns about the incremental nature of the contributions with insufficient evaluation on long-term generation. The authors have provided detailed rebuttals addressing cache sharing, cyclic TPE, and additional quantitative evaluations, so the AC leans toward accepting this submission.